# Particle swarm optimization algorithm-based PI inverter controller for a grid-connected PV system

**M. F. Roslan**[1]*, **Ali Q. Al-Shetwi**[2,3], **M. A. Hannan**[1], **P. J. Ker**[2], **A. W. M. Zuhdi**[2]

**1** Department of Electrical Power Engineering, Universiti Tenaga Nasional, Kajang, Selangor, Malaysia,
**2** Institute of Sustainable Energy, Universiti Tenaga Nasional, Kajang, Selangor, Malaysia, **3** Electrical Engineering Department, Fahad bin Sultan University, Tabuk, Saudi Arabia

* Mfirdaus@uniten.edu.my

**Data Availability Statement:** All relevant data are within the paper and its files.

## Abstract

The lack of control in voltage overshoot, transient response, and steady state error are major issues that are frequently encountered in a grid-connected photovoltaic (PV) system, resulting in poor power quality performance and damages to the overall power system. This paper presents the performance of a control strategy for an inverter in a three-phase grid-connected PV system. The system consists of a PV panel, a boost converter, a DC link, an inverter, and a resistor-inductor (RL) filter and is connected to the utility grid through a voltage source inverter. The main objective of the proposed strategy is to improve the power quality performance of the three-phase grid-connected inverter system by optimising the proportional-integral (PI) controller. Such a strategy aims to reduce the DC link input voltage fluctuation, decrease the harmonics, and stabilise the output current, voltage, frequency, and power flow. The particle swarm optimisation (PSO) technique was implemented to tune the PI controller parameters by minimising the error of the voltage regulator and current controller schemes in the inverter system. The system model and control strategies were implemented using MATLAB/Simulink environment (Version 2020A) Simscape-Power system toolbox. Results show that the proposed strategy outperformed other reported research works with total harmonic distortion (THD) at a grid voltage and current of 0.29% and 2.72%, respectively, and a transient response time of 0.1853s. Compared to conventional systems, the PI controller with PSO-based optimization provides less voltage overshoot by 11.1% while reducing the time to reach equilibrium state by 32.6%. The consideration of additional input parameters and the optimization of input parameters were identified to be the two main factors that contribute to the significant improvements in power quality control. Therefore, the proposed strategy effectively enhances the power quality of the utility grid, and such an enhancement contributes to the efficient and smooth integration of the PV system.

## 1. Introduction

Global warming, climate change, and alarming levels of air pollution are major concerns that affect human health and natural wealth. The cost of not addressing these issues today would

**Funding:** This work was funded by the Ministry of Higher Education, Malaysia under Universiti Tenaga Nasional using grant no. 20190101LRGS.

**Competing interests:** The authors have declared that no competing interests exist.

far exceed the expense of addressing it in the future. Thus, global energy investments have been shifted towards cleaner options, such as renewable energy. Photovoltaic (PV) energy, which is a renewable energy source, has a strong potential to meet the current and future energy demand and promises positive environmental impacts.

PV panels that have been widely used in power system applications are adopting distributed generation (DG) technologies, because they can operate independently from the larger grid. This DG unit has been developed widely in many countries, such as Japan, Germany, Spain, and USA [1]. However, power quality, stability, and power mismatch are major challenges in a PV system integrated to the grid through an inverter. An inverter is a power electronic device, which is used to convert a renewable direct current (DC) source into an alternating current (AC) source for providing power to AC loads or transporting power to the utility grid [2]. The proper control strategy of the inverter on the utilization of PV power is necessary as the PV power system becomes valuable and significant to the user, especially where utility is unavailable. In such a system, the current control strategy is responsible for ensuring the quality of the power supply from the PV system, which is connected to the grid.

One method used in the strategy is to implement a proportional-integral (PI) controller. In general, the PI controller has been widely used in many devices for various applications due to its simple control structure, easy design, low cost, and reliable performance [3, 4]. It can solve many problems in different applications, including high overshoot, steady-state error, and oscillation due to the variation in the system. The PI controller parameters must be obtained properly to ensure proper function and robust control of the control loop system [5]. Standard tuning methods, including Ziegler–Nichols, Cohen–Coon, and self-tuning fuzzy PI-based M-constrained integral gain optimisation (MIGO), can be used to find the best values for PI controller parameters [6, 7]. However, the insufficient process information involving mathematical model and trial and error leads to reduced robustness to the controller and unsatisfactory results. Sliding mode controllers (SMC) have also been introduced to solve periodic error in the inverter output voltage under load variation. The results were obtained with reduced THD for linear and non-linear load [8]. Many optimization techniques, such as genetic algorithm (GA), particle swarm optimization (PSO), differential evaluation (DE), ant colony optimization, neural-fuzzy logic, and many more, have also been implemented to improve the PI controller performance and regulate its parameters [9–11]. One study applied the GA in the back propagation neural network of the PID controller to optimize the initial weight and minimize the DC current [12]. The GA optimization shows better performance in the DC current suppression process by spending a long time in the optimization process, thus making it difficult to achieve an online optimization process. However, the PSO algorithm can achieve an optimal solution within a shorter computation time period than the GA technique as they have a single solution search space. Furthermore, another study used the ant colony algorithm (ACA) to optimize the PI parameters in the stand-alone PV system [13]. The Jaya algorithm optimization is also introduced to tune the PID controller for minimizing frequency deviation in hybrid electric vehicle (HEV) applications [14]. One study implemented the Jaya algorithm optimization in the parameter estimation of the soil water retention to validate the effectiveness of the algorithm [15]. Similarly, the BAT algorithm has been utilized in another work [16] to search for the optimal parameters of the PI controller in a grid-connected PV system. The PSO algorithm is widely used in the optimization solution because of its advantages, including strong robustness and global convergence capability, easy implementation, and simple calculation [17, 18]. Moreover, the PSO algorithm achieved an optimal solution within a shorter computation time than the GA technique, as the former has a single solution search space. As demonstrated in these studies, optimization algorithms play significant roles in achieving the optimal design of PI controllers for power converters in DG systems.

Therefore, the PSO is used in the current study for the optimal design of PI controller parameters. Its aim is to obtain the best optimum values of $K_p$ and $K_i$ in real-time operation, which improves the power quality and stability of the three-phase grid-connected PV inverter system. The proposed controller scheme is implemented using a synchronous reference frame with two PI controllers. The feed-forward compensation is applied to the inner current control loop of the inverter to achieve high dynamic response. In view of the above discussion, the references mentioned above have limitations in terms of insufficient process information, less robustness, longer computation time than PSO, facing parameter complexities and coding difficulties to find the best fitness value. Moreover, they did not sufficiently discuss the effects of inverter optimization to improve the power quality performance, reduce the DC link voltage fluctuation, stabilise the output current and voltage, smooth the power flow, decrease the harmonics, and stabilise the frequency in the grid-connected PV inverter. Therefore, this work aims to improve the power quality performance of the inverter system by reducing transient response, minimising time overshoot, and obtaining low steady-state error due to variation in loads as well as DC link stabilization. In addition, the PSO is used to obtain the optimum values of the PI control parameters and reduce error to enhance the voltage and frequency stability with shorter time and less complexity.

The implementation of the PSO algorithm is achieved through the use of the (m-file) code MATLAB, whilst MATLAB/Simulink is used to design the inverter control scheme using a PV power generation of 100 kW. The simulated results demonstrate that the PI controller integrated with optimisation method provides excellent performance with low variation in total harmonic distortion (THD) compared to the PI controller without optimisation technique. The contribution and key highlights of the paper are as follows:

i. It proposes an optimized controller-based PSO algorithm to obtain the optimum values of $K_p$ and $K_i$ in real-time operation to improve the power quality and stability of the three-phase grid-connected PV inverter system.

ii. It solves the complicated constraints of the optimization problem, while obtaining an optimum value of the PI controllers' parameters by reducing transient response, minimising time overshoot, and obtaining low steady-state error due to variation in loads conditions.

iii. It offers a real-time operation of the PV inverter system with optimized controller by minimizing the error as much as possible, thus finding the best optimum parameters of PI controllers under load variation condition.

## 2. Description of the grid-connected PV inverter system

The grid-connected PV system with a three-phase voltage source inverter (VSI) used in this study is illustrated in Fig 1. It includes a PV system, maximum power point tracking (MPPT) algorithm, an inverter, a filter, and load. The DC link capacitors are connected to the output terminal of the PV system, and the output becomes the input voltage of the inverter system. The boost converter and MPPT algorithm are used to extract the maximum available power from the PV system. The three-phase VSI with its control and RLC filter is connected to the low-voltage AC grid through a step-up transformer on the distribution side to supply load. A detailed dynamic model of the grid-connected PV system will be described below.

### 2.1 PV system model

The fundamental part of a PV system is called the solar cell. Multiple individual solar cells connected in series will form a PV module [17, 18]. The equivalent circuit of a PV module,

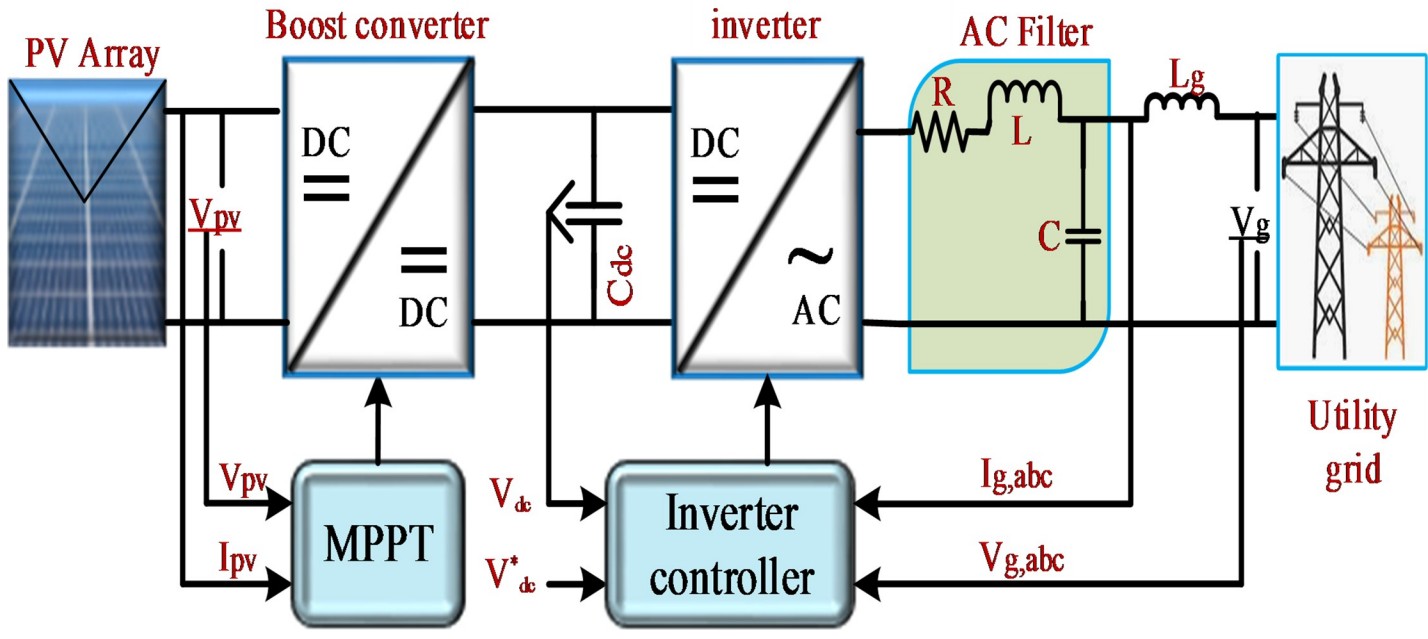

**Fig 1. Schematic of the grid-connected PV system.**

illustrated in Fig 2, consists of a photocurrent source $I_{ph}$, a parallel diode to the source (D), series resistance $R_s$, and shunt resistance $R_{sh}$ arranged in parallel with the diode.

The relationship between the output voltage and current based on the equivalent circuit is expressed as in Eq (1) [18, 19]:

$$I_{pv} = I_{ph} - I_{sh}\left(exp\left(\frac{V_{pv} + IR_s}{AN_s kT}\right) - 1\right) - \frac{V_{pv} + IR_s}{R_{sh}}, \tag{1}$$

where $I_{sh}$ represents the photocurrent, $I_{sh}$ is the saturation current, $A$ is the ideal diode factor, $q$ is the electron charge ($1.602\times10^{-19}$ C), $k$ is the Boltzmann's constant ($1.381\times10^{-23}$), $T$ is the cell temperature, and $N_s$ is the number of PV cells connected in series. The photocurrent of the PV cell at any solar irradiation and temperature can be computed using Eqs (2), (3) and (4)

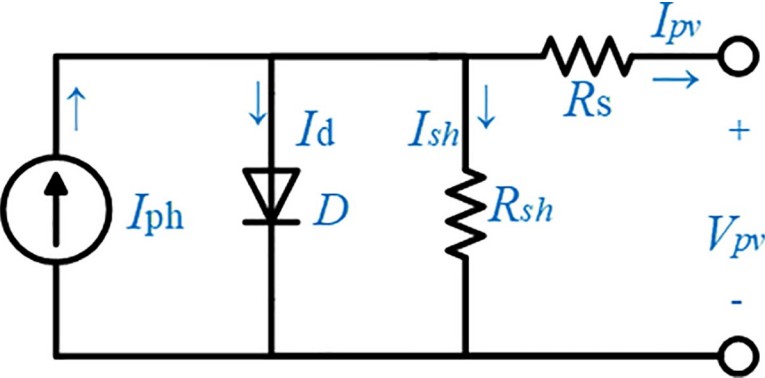

**Fig 2. Equivalent circuit of the PV model.**

**Table 1. Main parameters of the PV module.**

| Parameters of the PV module | Value |
|---|---|
| Maximum power | $P_{max}$ = 400 W |
| Maximum power voltage | $V_{mp}$ = 54.7 V |
| Maximum power current | $I_{mp}$ = 5.58 A |
| Open-circuit voltage | $V_{oc}$ = 64.2 V |
| Short-circuit current | $I_{sc}$ = 5.96 A |
| Cell number per module | $N_s$ = 66 |
| Temperature coefficient of $I_{sc}$ | $K_i$ = 0.062/˚C |
| Temperature coefficient of $V_{oc}$ | $K_v$ = −0.273/˚C |
| Ideality factor of the diode | A = 0.95 |

[19, 20]

$$I_{ph} = \frac{G}{G_{ref}} * [I_{sc} + K_i(T - T_{rk})], \tag{2}$$

$$I_{sc} = I_{sc,ref}\left(\frac{R_p + R_s}{R_p}\right), \tag{3}$$

$$I_{sat} = \frac{I_{SC,ref} + K_i(T - T_{rk})}{e^{q\left(V_{oc,ref} + \frac{K_v(T-T_{rk})}{AN_sKT}\right)} - 1}, \tag{4}$$

where $G_{ref}$ and $G$ are the nominal and actual solar irradiation, respectively; $T_{rk}$ is the module absolute temperature in Kelvin; $K_i$ is the temperature coefficient of short-circuit current; and $K_v$ is the temperature coefficient of open-circuit voltage. In addition, $I_{SC,ref}$ and $V_{oc,ref}$ are the short circuit current and open circuit voltage of the module at a standard test condition, respectively [21]. The parameter values of the PV module based on Eqs (1)–(4) are described in Table 1. Based on the above-mentioned mathematical model of the PV module, the P–V and I–V characteristic curves are verified through Simulink model under the standard test condition with $G_{ref}$ = 1000 W/m² and $T_{rk}$ = 25˚C, as shown in Fig 3.

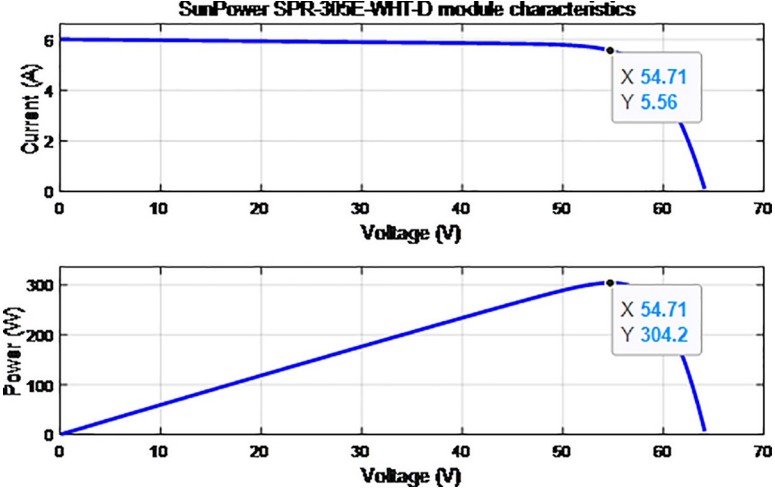

**Fig 3. Characteristic curves of PV.**

## 2.2 DC–DC boost converter and MPPT algorithm

The proposed PV array above is selected and modelled to produce 100 kW of peak power with 66 series modules and 5 parallel strings. Fig 3 shows the characteristic curves of the PV system array with different levels of irradiation and temperature. The nonlinear variation in solar irradiance and temperature, which considerably affects the voltage and current of the PV array, tends to produce inconstant available power [22]. Thus, the MPPT method, namely, perturb and observe (P&O), is used in this system together with DC–DC boost converter to track the maximum power produced by PV array based on the irradiation and temperature levels. Details of the MPPT method and algorithm are further discussed in [23, 24]. The P&O technique is chosen in this proposed system because of its simplicity and feasibility in sensing the PV voltage $V_{pv}$ and PV current $I_{pv}$ to regulate the maximum PV power. By changing the duty cycle $D$ of the PWM devices in the boost converter, the extracted PV voltage will increase to 500 V for the DC voltage $V_{dc}$. A DC link is widely used in the grid-connected PV systems. The DC link should have a value near the maximum array voltage in order to reduce the output current ripple as well as regulate and stabilize the voltage at the DC side of the grid inverter. The inverter has the same nominal voltage as the DC link and should be compatible with the array current and voltage to withstand its maximum values within the inverter voltage range. The mathematical equation of the boost converter is shown in Eq (5).

$$V_{dc} = \frac{V_{pv}}{1-D} \tag{5}$$

## 2.3 DC–AC inverter

An inverter is a device that converts a DC source into an AC source through an AC filter, which reduces high-frequency harmonics to the grid system [2]. The PWM logic signals for the inverter switch are generated by the controller according to the control strategy and inverter output parameter. A block diagram of a normal operation for an inverter control system is depicted in Fig 4. It mainly consists of the DC link voltage, the insulated gate bipolar transistor (IGBT), and filters. The input DC link consists of a 1200 μF capacitor that links the DC power to the inverter system to stabilize the input voltage to be injected to the inverter. The inverter is connected to the grid through an AC filter to reduce the high-frequency harmonic component injected into the grid system [25]. The control system of the inverter consists of an inner current loop and voltage control loop. The PI controller in the inner current loop helps to regulate the grid current and DC link voltage while stabilizing and managing the DC link voltage $V_{dc}$ in the voltage loop [19, 24].

## 3. Inverter control strategy

The grid voltage and phase angle are synchronized through the phase-lock loop (PLL) based on the direct-quadrature (*d-q*) synchronous references frame using Park's Transformation. The regulation in the *d-q* domain is easy and has good dynamic response [27]. The reference angle is obtained by integrating the reference angular frequencies. The phase angle $\theta$ obtained from the PLL will be used in *abc* to *d-q* transformation for the grid voltage $V_{g,abc}$ and grid current $I_{g,abc}$ to produce voltage and current in the *d-q* axes reference frame, respectively. The measured at the PCC is transformed to the *d-q* axes references frame using Eqs (6) and (7) [28]

$$\begin{pmatrix} V_{od} \\ V_{oq} \\ V_0 \end{pmatrix} = \sqrt{\frac{2}{3}} \begin{pmatrix} cos\theta & \cos(\theta - 2\pi/3) & \cos(\theta + 2\pi/3) \\ -sin\theta & -\sin(\theta - 2\pi/3) & -\sin(\theta + 2\pi/3) \\ 1/2 & 1/2 & 1/2 \end{pmatrix} \begin{pmatrix} V_{ga} \\ V_{gb} \\ V_{gc} \end{pmatrix}, \tag{6}$$

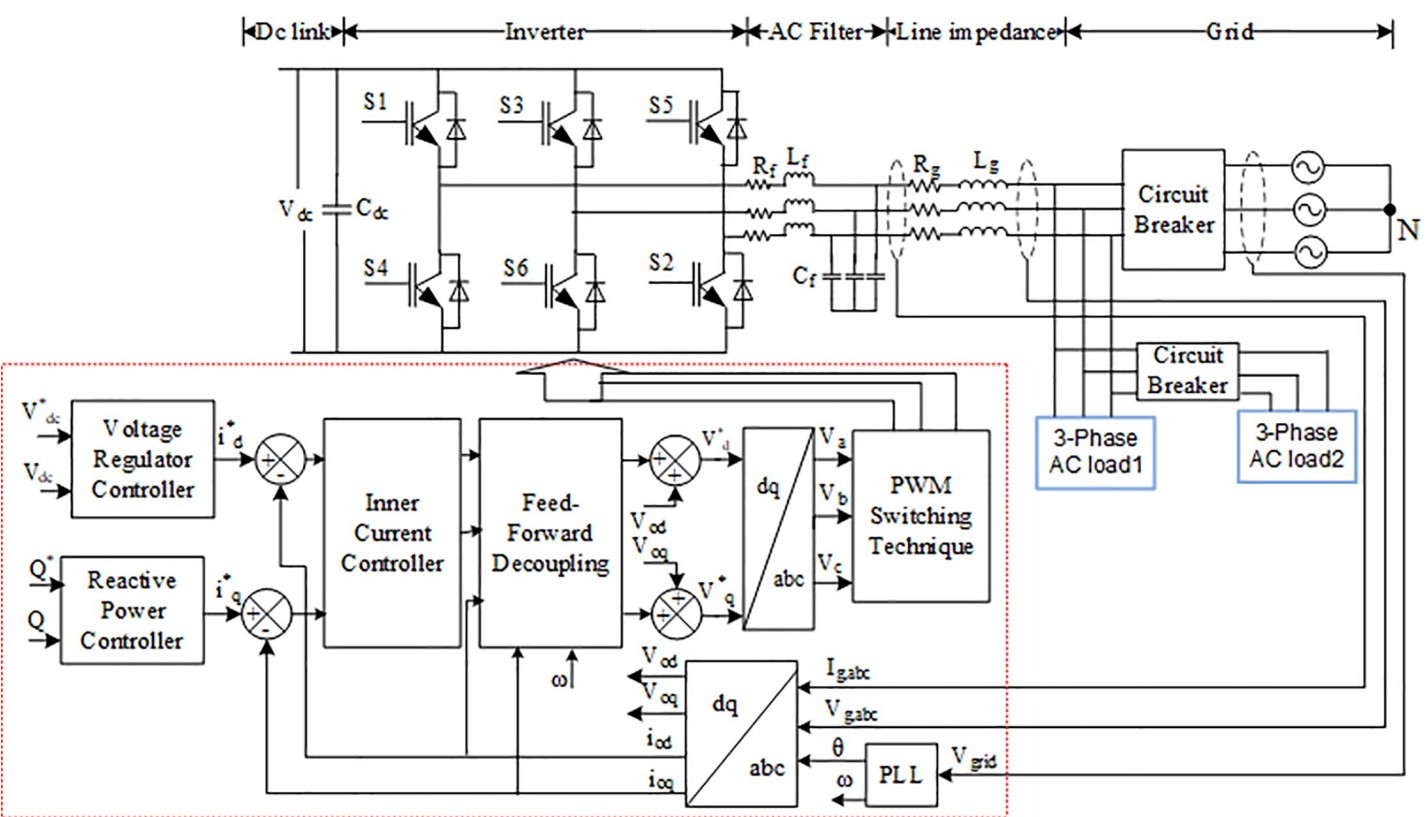

**Fig 4. Grid-connected PV system with three-phase inverter control scheme.**

$$\begin{pmatrix} i_{od} \\ i_{oq} \\ i_0 \end{pmatrix} = \sqrt{\frac{2}{3}} \begin{pmatrix} cos\theta & cos(\theta - 2\pi/3) & cos(\theta + 2\pi/3) \\ -sin\theta & -sin(\theta - 2\pi/3) & -sin(\theta + 2\pi/3) \\ 1/2 & 1/2 & 1/2 \end{pmatrix} \begin{pmatrix} I_{ga} \\ I_{gb} \\ I_{gc} \end{pmatrix}, \tag{7}$$

where $V_{od}$, $i_{od}$ and $V_{oq}$, $i_{oq}$ are the voltage and current in the $d$–$q$ axis reference frames, respectively, and ($V_{ga}$, $V_{gb}$, and $V_{gc}$) and ($I_{ga}$, $I_{gb}$, and $I_{gc}$) are the three-phase voltage and current of the grid, respectively. The $abc$ to $d$-$q$ transformation is essential in inverter operation as the inner controller of inverter control works in the $d$-$q$ model in DC value asymmetrical system, which is done by the PI controller. The PI controller in the inner loop control will produce a reference active voltage $V_d^*$ and reference reactive voltage $V_q^*$, which will be used for the $d$-$q$ to $abc$ transformation in order to produce $V_{abc}$. A feed-forward term should be used to decouple the active and reactive current axes in the inner loop control [29]. The values of $V_{abc}$, are then sent to the PWM signal to generate a control gate pulse for the inverter.

After applying Park's transformation, the real and reactive power can be calculated using Eqs (8) and (9) [19, 26]

$$P = \frac{3}{2} \left( V_{od} * i_{od} + V_{oq} * i_{oq} \right), \tag{8}$$

$$Q = \frac{3}{2} \left( V_{oq} * i_{od} - V_{od} * i_{oq} \right), \tag{9}$$

where $V_{od}$, $V_{oq}$ and $i_{od}$, $i_{oq}$ are respectively denoted as voltage and current in the $d$-$q$ axes reference frames on the grid side. The current control in the inner control loop in the inverter controller scheme accurately tracks the current signal and removes short-circuit current transient. From the inverter inner current loop control shown in Fig 4, the output active and reactive voltage ($V_d^*$ and $V_q^*$) of the grid inverter in the $d$-$q$ synchronous frame at the line frequency can be respectively described using Eqs (10) and (11).

$$V_d^* = i_d^* - i_{od}\left(k_{pd} + \frac{k_{id}}{s}\right) - w * L_f * i_{oq} + V_{od} \tag{10}$$

$$V_q^* = i_q^* - i_{od}\left(k_{pq} + \frac{k_{iq}}{s}\right) + w * L_f * i_{od} + V_{oq} \tag{11}$$

The controlled pulse from the inner loop control with the controlled amount of power is then used for switching the VSI inverter. In turn, this will be injected into the power system to produce excellent power quality. The main parameters of the grid connected inverter system are listed in Table 2. More information on the grid connected inverter control system can be found in [27, 28]. The controlled pulse from the inner loop control with controlled amount of power is then used to switch the VSI inverter and then injected into the power system to produce excellent power quality. The main parameters of the grid-connected inverter system are listed in Table 2. Additional information on the grid-connected inverter control system can be found in [27, 28].

## 4. System modelling

### 4.1 Phase-Lock Loop (PLL)

The PLL system indicated in Fig 4 synchronizes the utility network to provide an accurate and fast detection of the utility phase angle in the grid-connected inverter [27]. Fig 5 shows the block diagram of the synchronous PLL frame. The PI regulator is used to drive $V_{oq}$ to be zero and minimize error to set the rotation frequency. The phase angle $\theta$ is obtained by integrating the angular frequency, which can be expressed as follows [29, 30]:

$$\omega = K_P^{PLL} V_{oq} + K_I^{PLL} \int_0^t V_{oq} dt, \tag{12}$$

$$\theta = \int_0^t \omega \, dt. \tag{13}$$

Table 2. Main parameters of the inverter connected grid.

| Parameters of the inverter | Value |
|---|---|
| Effective voltage of the grid | $V_{grid}$ = 33 kV |
| DC link voltage | $V_{dc}$ = 500 V |
| DC link capacitor | $C_{dc}$ = 1200 μF |
| Grid frequency | ω = 2π*50 rad/s |
| R filter of the inverter | $R_f$ = 1.89 mΩ |
| L filter of the inverter | $L_f$ = 250 μH |
| Switching frequency of the inverter | $f_s$ = 2 kHz |
| Line Resistance | $R_g$ = 0.04 Ω/km |
| Line Impedance | $X_g$ = 0.13 Ω/km |
| Load 1 | $P_{Load1}$ = 100 kW |
| Load 2 | $P_{Load2}$ = 30 kW |

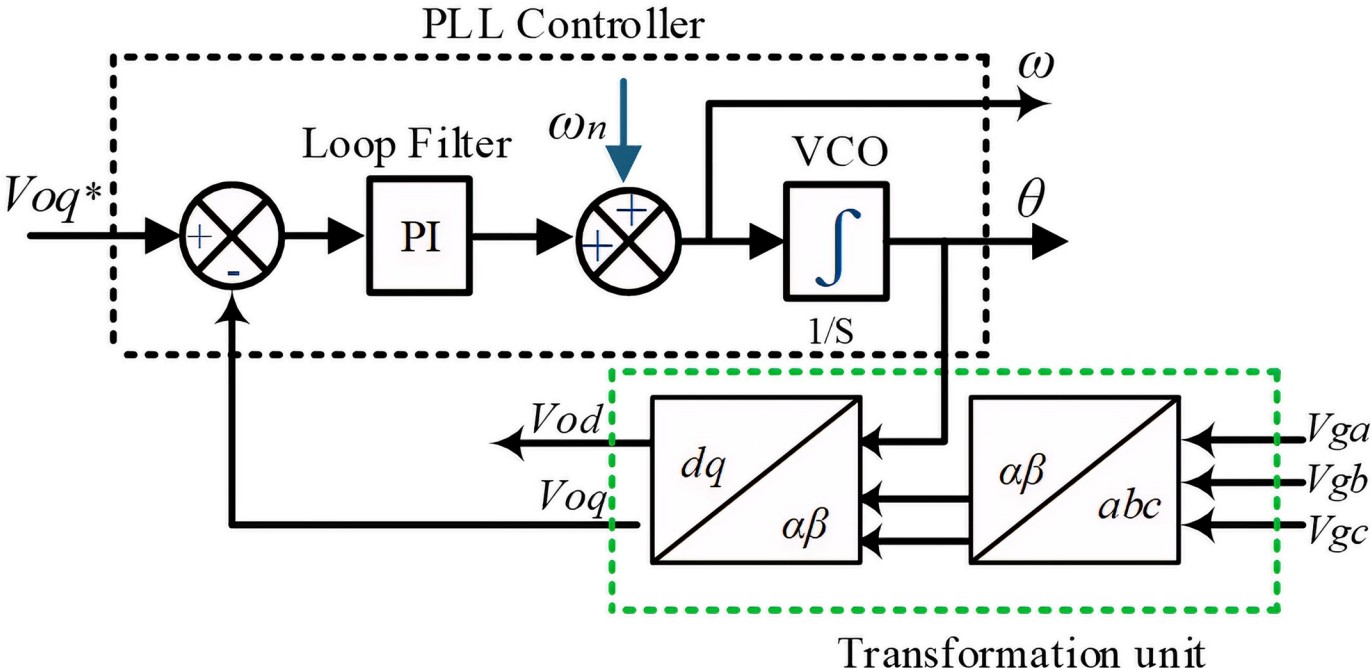

**Fig 5. Block diagram of the PLL.**

## 4.2 Voltage regulator controller

The DC link is a crucial part of the grid-connected PV system and acts as the DC bus between the boost converter and inverter. The DC link capacitor is an entity used to improve power quality and protection and to meet the requirements of the grid system [21]. By using this DC link regulator, the power flow in the grid system can be regulated. The PI controller is used in order to generate a reference peak current $i_d^*$ for the grid. Therefore, the output of the outer voltage controller is the input of the inner controller loop. The model of the DC link voltage regulator is expressed as Eq (14)

$$i_d^* = e * (K_{pdv} + \int K_{idv}), \qquad (14)$$

where e is the difference between the reference and measured voltage. The calculated error will become an input for the PI controller to maintain a DC link voltage with minimum error. A constant DC link voltage should be maintained between the boost converter and the inverter device, because any fluctuation across the DC link will cause THD, which leads to poor power quality in the grid system [31].

## 4.3 Current controller

The current controller in grid-connected mode is essential for power quality improvement. As shown in Fig 4, the current controller is implemented with a feed-forward decoupling strategy for connection voltage by eliminating the current error. The PLL should be used in this control strategy in detecting the grid voltage phase angle in order to implement Park's transformation according to Eqs (6) and (7) for the current control scheme in the *d-q* synchronous frame [32]. The current controller block diagram is shown in Fig 6.

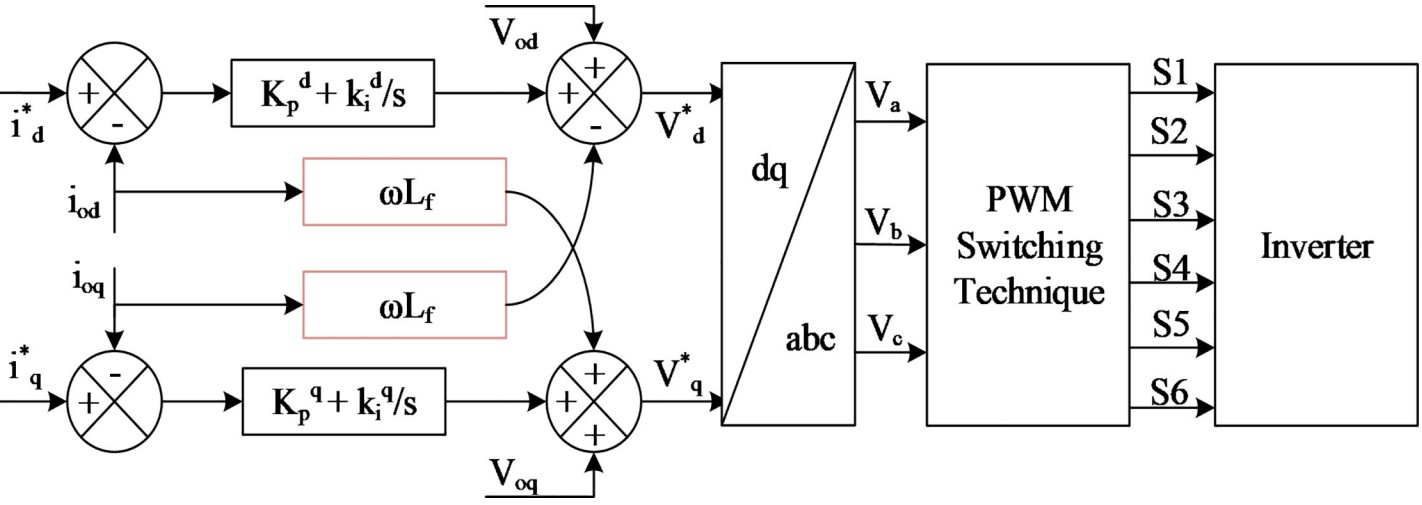

**Fig 6. Current controller in the grid-connected mode.**

From Fig 6, the following equation can be obtained:

$$\begin{bmatrix} V_d^* \\ V_q^* \end{bmatrix} = \begin{bmatrix} -K_p^d & -\omega L_f \\ \omega L_f & -K_p^d \end{bmatrix} \begin{bmatrix} i_{od} \\ i_{oq} \end{bmatrix} + \begin{bmatrix} K_p^d & 0 \\ 0 & K_p^d \end{bmatrix} \begin{bmatrix} i_d^* \\ i_q^* \end{bmatrix} + \begin{bmatrix} K_i^d & 0 \\ 0 & K_i^d \end{bmatrix} \begin{bmatrix} x_d \\ x_q \end{bmatrix} + \begin{bmatrix} V_{od} \\ V_{oq} \end{bmatrix}, \quad (15)$$

$$\text{where} \qquad x_d = (i_d^* - i_{od})/s, \qquad (16)$$

$$x_q = (i_q^* - i_{oq})/s. \qquad (17)$$

The output signal from the current controller is converted from the *d-q* synchronous references frame to *abc* transformation for feeding into the pulse width modulation (PWM) technique and switching the inverter.

## 5. PI-based PSO algorithm

In general, the PI controller is a feedback control loop that is designed based on a linearized system equation for the load conditional-based operations [33]. Good control performance, easy implementation, and high reliability are the main factors for choosing the PI controller for controlling an inverter system. Most of the AC system voltage and current regulators utilize the PI controller to regulate the grid current and DC link voltage, thus properly controlling the active power injected to the grid system and stabilizing the DC link voltage of the inverter system. This is because the huge impact of the nonlinearity behaviour of the inverter and loads causes the voltage and current harmonics of the system, which in turn, leads to the destruction of the overall power system [34]. In this regard, many studies dealt with the PI controller design for grid-connected inverters using an inductive filter. For instance, the authors in [35] have proposed a simple step-by-step controller design method for the LCL-type grid-connected inverter. By carefully dealing with the interaction between the current regulator and active damping, the complete satisfactory regions of the controller parameters for meeting the system specifications are obtained, from which the controller parameters can be easily picked out. Moreover, the stability problem of the grid-connected inverter with LC filters is annualized in [36]. The results of that work demonstrated that the possible grid-impedance variations have a significant influence on the system stability when a conventional PI controller is used

for grid current control. Conventionally, the PI controller has been employed in synchronous reference frames due to its simplicity and stability. Although the PI controllers are relatively straightforward to tuning, the variation in the nonlinear load condition and grid disturbances leads to significant challenges in achieving an optimal and fully robust solution and depends on the designer for obtaining the best performance [37]. A recent research has proven that a control system with a PI controller using fractional order implemented in a three-phase inverter system can mitigate poor voltage regulation in a grid-connected PV system [38]. Moreover, the finite model predictive control using the L-type filter, which is also implemented in a grid-connected inverter system, has been found to mitigate current harmonic [39]. However, this control approach requires the measurement of all system state variables, which brings about more complexity for the inverter system. In the current study, PSO algorithm optimization technique is used for the optimal design of the PI controller parameters for obtaining the best optimum values of $K_p$ and $K_i$ in real-time operation to reduce transient response, minimize time overshoot, and obtain low steady state error due to load variations in the three-phase, grid-connected PV inverter system. The purpose is to ensure that an optimization technique with PSO algorithm can work properly and provide fast response by searching the optimum values of $K_p$ and $K_i$ in real-time operation. This PSO algorithm optimization technique will respond immediately to address the input error of the system. Fig 7 shows a proposed model of the inverter control system with the PSO technique. A PSO technique is implemented to optimally design the PI controller parameters by minimizing the error between the voltage regulator and current controllers. Detailed descriptions of the fitness function and constraints in the PSO algorithm are provided below. The main advantages and disadvantages of the conventional Ziegler-Nichols technique, fuzzy logic controller, neural fuzzy logic, GA, and PSO algorithm are described in Table 3. From the table, we can see that the PSO algorithm has a simple set of parameters during implementation, which requires a shorter time to converge to an optimum solution compared to other techniques. Furthermore, the PSO algorithm does not require inferences rules during implementation compared to the fuzzy logic controller and neural fuzzy logic, which requires a time-consuming trial and error process. Therefore, the

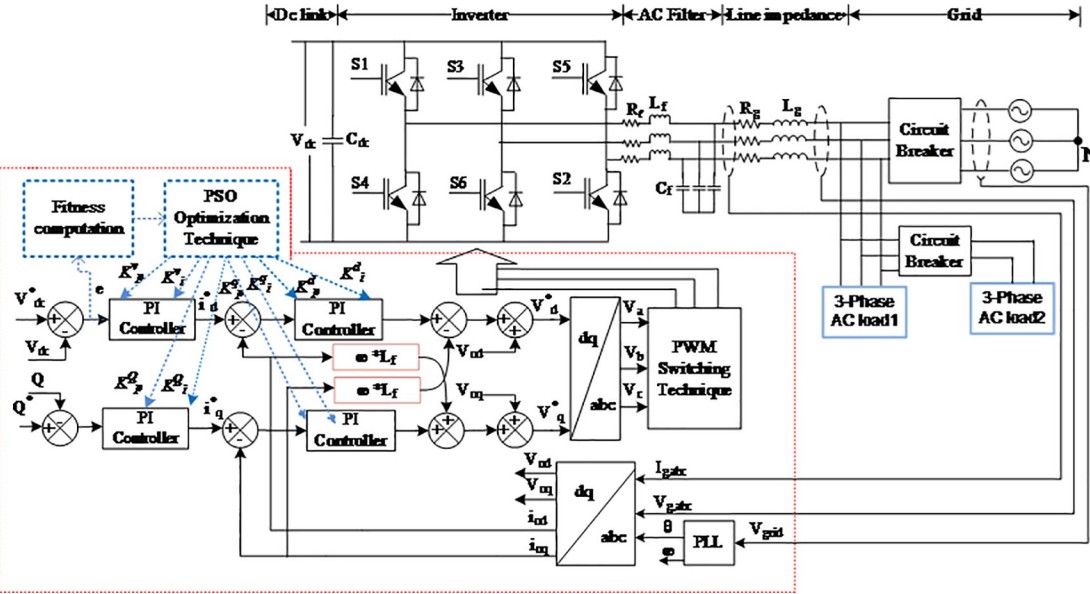

**Fig 7. Schematic diagram of the proposed PSO optimization technique for the inverter control scheme.**

**Table 3. Advantages and disadvantages of the control algorithm.**

| Algorithm | Data Training | Rules | Function evaluation | Parameter setting | References |
|---|---|---|---|---|---|
| Ziegler–Nichols | No | No | No | Simple with calculation | [6] |
| Fuzzy Logic controller | No | Yes | No | Simple but trial and error | [7] |
| Neural Fuzzy Logic | Yes | Yes | No | Huge set of parameters | [9, 11] |
| GA | No | No | Yes | Huge set of parameters | [12] |
| PSO | No | No | Yes | Simple set of parameters | [17, 18] |

PSO algorithm leads to a simple and fast implementation process with single solution space to obtain the optimum values of the PI controller parameters with shorter time responses.

## 5.1 Fitness function

Fitness function is the optimal solution of evaluating the searching area called objective function. The mean square error and the THD are formulated as objective functions to be minimised for the DC link voltage and current controller, in order to obtain the optimal PI controller parameters $K_p$ and $K_i$ in the DC link voltage regulator and current controller [36]. The objective function of the system can be described as follows:

$$min\, F(x) = C_1 \int_0^{T_{max}} t|e(t)|dt + C_2 \int_0^{T_{max}} THD_v\, dt, \tag{18}$$

where $e$ is the error, $C_1$ and $C_2$ are weight coefficients, $T_{max}$ is the maximum time, and $THD_V$ is the THD of output voltage.

## 5.2 Problem constraints

In the grid connected-mode of a PV system, the problem constraints are the optimized parameters containing eight parameters of decoupled PI controllers, namely,
$K_P^v,\ K_i^v,\ K_P^Q,\ K_i^Q K_P^d,\ K_i^d,\ K_P^q,$ and $K_i^q$. The complete formulation of the optimal DC link voltage regulator and current controller for the three-phase grid-connected inverter of the PV system is expressed below.

$$min\, F(x) = C_1 \int_0^{T_{max}} t|e(t)|dt + C_2 \int_0^{T_{max}} THD_v\, dt \tag{19}$$

$$l_1 \leq K_P^v \leq u_1$$

$$l_2 \leq K_i^v \leq u_2$$

$$l_3 \leq K_P^Q \leq u_3$$

$$l_4 \leq K_i^Q \leq u_4$$

$$l_5 \leq K_P^d \leq u_5$$

$$l_6 \leq K_i^d \leq u_6$$

$$l_7 \leq K_P^q \leq u_7$$

$$l_8 \leq K_i^q \leq u_8$$

$$x = (K_P^v, K_i^v, K_P^Q, K_i^Q, K_P^d, K_i^d, K_P^q, K_i^q)$$

In the expressions above, $l_1$, $l_2$, $l_3$, $l_4$, $l_5$, $l_6$, $l_7$, and $l_8$ are the lower limits of $K_P^v, K_i^v, K_P^Q, K_i^Q, K_P^d, K_i^d, K_P^q$, and $K_i^q$, respectively, and $u_1$, $u_2$, $u_3$, $u_4$, $u_5$, $u_6$, $u_7$, and $u_8$ are the upper limits of $K_p^v$, $K_i^v, K_P^Q$, $K_i^Q$ $K_P^d$, $K_i^d$, $K_P^q$, and $K_i^q$, respectively. In this study, the PSO algorithm is proposed to solve this optimization problem.

## 5.3 PSO algorithm

In this section, the proposed PSO method is first introduced. Then, the implementation of the technique in this work is discussed. The evolutionary algorithm called PSO was first proposed by Elberhat and Kennedy in 1995 to solve various real-valued optimization problems [15, 40]. PSO is a powerful technique for achieving the best solution in a nonlinear system. The PSO technique is developed on the basis of the natural behaviours of flocks of birds and schools of fishes that move around in groups at a D-dimensional space [37]. At every iteration, each particle moves in the direction of the best solution discovered so far in the swarm. Keeping this interaction, the particle continues searching for a better solution than the previous one and moves toward it, thereby exploring the region thoroughly [41, 42]. The position and velocity of the $i^{th}$ particle of the swarm in the search space vector are represented as $X_i = [X_{i1}, X_{i2}, \ldots, X_{iD}]$ and $V_i = [V_{i1}, V_{i2}, \ldots, V_{iD}]$, respectively [43]. The best previous solution for the $i^{th}$ particle swarm is $P_i = [P_{i1}, P_{i2}, \ldots, P_{iD}]$, and the global best position is $P_g = [P_{g1}, P_{g2}, \ldots, P_{gD}]$ [38]. The updated position and velocity of each particle in the next iteration are based on Eqs (20) and (21) [44, 45], respectively,

$$V_i^{n+1} = \omega V_i^n + C_1 r_1 (P_i^n - X_i^n) + C_2 r_2 (P_g^n - X_i^n), \qquad (20)$$

$$X_i^{n+1} = X_i^n + V_i^{n+1}, \qquad (21)$$

where ($i = 1, 2, \ldots, m$), $n$ is the iteration number, $\omega$ is the inertia weight, $C_1$ is the social rate, $C_2$ is the cognitive rate, and $r_1$ and $r_2$ are random intervals (0, 1). The PSO algorithm is used in this study due to its easy implementation, robustness, global convergence capability and minimal required computation time. Tuning the PSO parameters considerably affects the optimisation performance. Thus, choosing the proper parameters helps improve the algorithm's effectiveness. In the inverter control system, the best values of the $K_p$ and $K_i$ parameters will be identified to produce small or zero error.

## 5.4 PSO algorithm implementation

The PSO algorithm randomly initiates the particle at the beginning to calculate the fitness and obtain the best value of each parameters in the whole swarm. The proposed PSO algorithm was implemented using MATLAB code (Version 2019B) 2.50 GHZ PC with 8GB RAM. The flowchart of the PSO algorithm is shown in Fig 8. The optimization steps of the PSO algorithm are described below.

**Step 1:** Initialize the parameters: in this step, the parameters of the algorithm are defined, including the number of population (N), lower and upper bounds (LB, UB), maximum number of iteration (I), weight coefficients ($C_1$, $C_2$), velocity (V), inertia constant (w), and number of variable (NV).

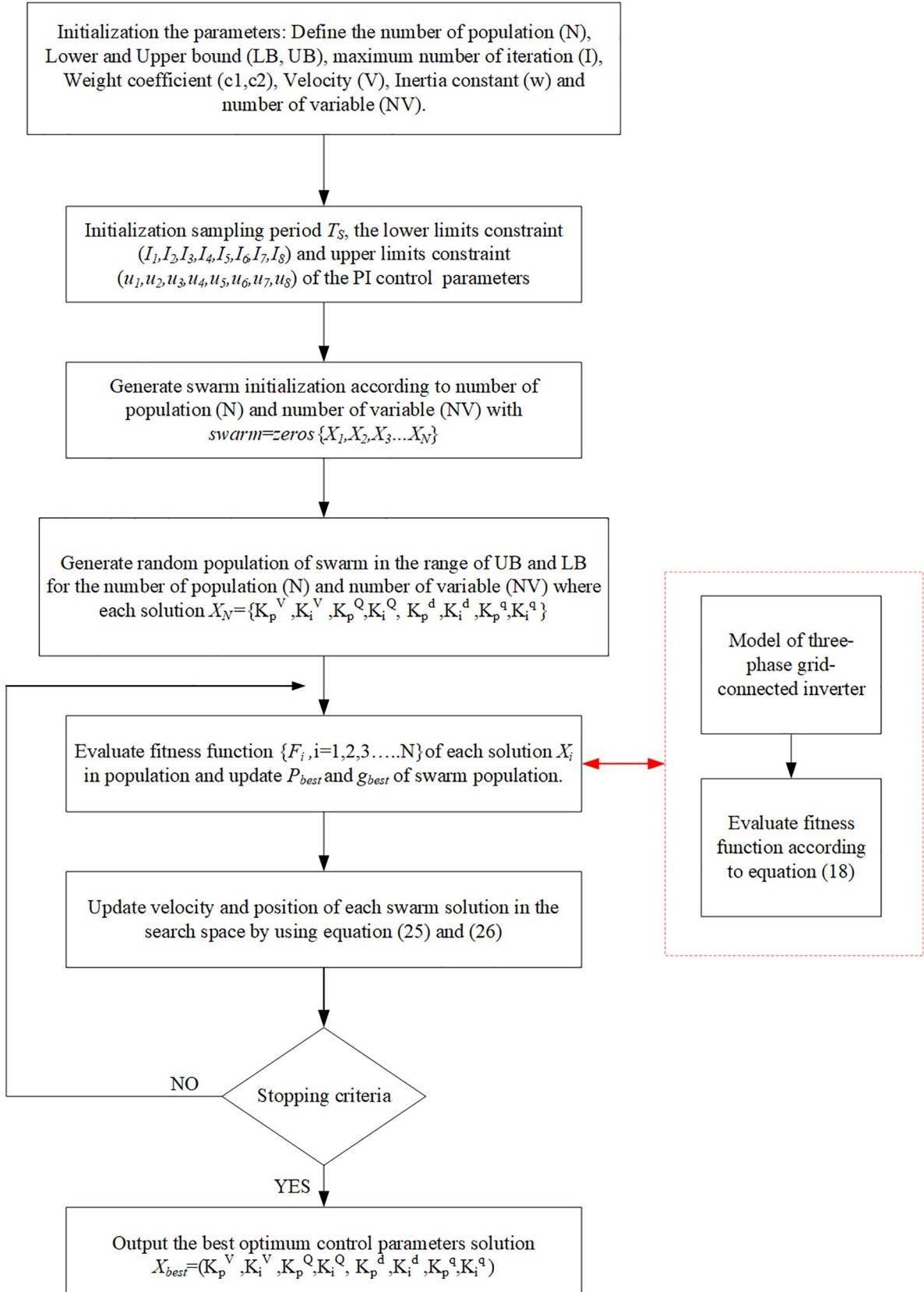

**Fig 8. Flowchart of the PSO algorithm for the three-phase grid-connected inverter control system.**

Step 1.1: Set the constraints: the lower limit constraints $(l_1, l_2, l_3, l_4, l_5, l_6, l_7, l_8)$ and upper limit constraints $(u_1, u_2, u_3, u_4, u_5, u_6, u_7, u_8)$.

Step 1.2: Define the control parameters: define the PI control parameter $(K_P^v, K_i^v, K_P^Q, K_i^Q, K_P^d, K_i^d, K_P^q,$ and $K_i^q)$ and initialize swarms as zero in the vector according to the number of population (N) and the number of variables (NV) as follows:

$$\text{Swarm} = \text{zeros} \begin{bmatrix} X_{1,1} & X_{1,2} & \cdots & X_{1,NV} \\ X_{2,1} & X_{2,2} & \cdots & X_{2,NV} \\ \vdots & \vdots & \ddots & \vdots \\ X_{N,1} & X_{N,2} & \cdots & X_{N,NV} \end{bmatrix}. \tag{22}$$

**Step 2:** Initialize the swarm population. In this step, the swarm population is created, whose matrix is equal to NV $^*$N, and is determined as follows:

$$X_{i,j} = \text{round}\,(LB_j + \text{rand}\,() * (UB_j - LB_j)) \tag{23}$$

$$i = 1, 2, \ldots, N, j = 1, 2, \ldots .NV$$

The initialization value of the population in the population vectors is as follows:

$$\text{Swarm} = \begin{bmatrix} X_{1,1} & X_{1,2} & \cdots & X_{1,NV} \\ X_{2,1} & X_{2,2} & \cdots & X_{2,NV} \\ \vdots & \vdots & \ddots & \vdots \\ X_{N,1} & X_{N,2} & \cdots & X_{N,NV} \end{bmatrix}. \tag{24}$$

Each swarm utilizes its memory and flies through the search space for obtaining a better position than its current one.

**Step 3:** Improve the particle generated by the new swarm. In its memory, a swarm memorizes the best experiences it has found $(P_{best})$ along with the group of best experience $(G_{best})$. In this step, a new swarm is generated as follows:

Step 3.1: the velocity of the swarm is updated as follows:

$$V_{i,j}^{t+1} = wV_{i,j}^t + C_1 rand[0,1] \text{ X } (Pbest_{i,j}^t - X_{i,j}^t) + C_2 rand[0,1] \text{ X } (Gbest - X_{(i,j)}^t). \tag{25}$$

Step 3.2: the position of the swarm is updated as follows:

$$X_{i,j}^{t+1} = V_{i,j}^{t+1} + X_{i,j}^t. \tag{26}$$

**Step 4:** Conduct fitness evaluation: The new vector is already improved according to the above rule and is evaluated based on fitness function. The personal and global best are updated as follows:

$$\text{If } fitness(X_{i,j}^{t+1}) < fitness(Pbest_{i,j}^{t}) \text{ then,}$$

$$Pbest_{i,j}^{t} = X_{i,j}^{t+1} \tag{27}$$

end

$$if \quad fitness(Pbest_{i,j}^{t}) < fitness(Gbest) \text{ then}$$

$$Gbest = Pbest_{i,j}^{t} \tag{28}$$

end

**Step 5:** Stopping criterion: If the number of iterations exceeds the maximum allowed, then the optimization is stopped. Otherwise, go to steps 3 and 4.

The proposed optimisation technique represented in the pseudocode, which shows how the developed PSO algorithm searches for the best space solution for the best position, is presented in Fig 9.

The PSO algorithm is used to search for the optimum values of the PI controller parameters in order to provide an effective controller and better switching state for an inverter. The optimum parameters of the PSO algorithm are identified as follows: number of population (N) is 50, maximum number of iteration (I) is 100, weight coefficient ($C_1$ and $C_2$) is 2, and inertia constant (w) is 1.

## 6. Results and discussion

A 100 KW grid-connected PV power is implemented in the MATLAB/Simulink with constant irradiance and temperature at 1000 $W/m^2$ and 25˚$C$, respectively, to implement the proposed PSO for tuning the PI controller parameters in an inverter control scheme. The simulation and test results are presented as follows for verifying the results of the proposed PSO for the inverter control scheme. The main highlights of the obtained results are the best optimum values of the PI controller parameters, fare comparison of results obtained without and with optimisation, and the THD analysis of the voltage. The latter shows how the proposed optimisation method improves the power quality of the three-phase grid-connected inverter for the PV system.

Table 4 compares the performance of the final PI controller parameters of the three-phase grid-connected PV system. These parameters were implemented in the control algorithm of the inverter controller to enhance the performance of the inverter system. The initial value initiated to the PI controller of the inverter system was implemented by using the trial and error phase. These trial and error values are initially implemented in the model of the three-phase grid-connected PV system in the MATLAB/Simulink environment simulation. Then, the PSO algorithm code was simulated, and the model was run simultaneously with 100 iterations. This process is repeated until reaching the maximum number of 100 iterations and provides the final values of the PI controller, as shown in the Table 4.

**Pseudocode of PSO-based PI Controller Algorithm**

**Input**: *I (maximum number of iteration), N (number of population size), NV (number of variable), D(number of dimension), $C_1, C_2$ (number of cognitive factor), step (number of step) ω (inertia constant), UB (upper bounds), LB (lower bound).*

**Output**: *globalBest Gbest, F(x) (fitness function).*

1. Initialize swarm positions and velocities randomly.
   for *i* from **1 to** *N* do
2. **X0(i,j)=round(LB(j)+rand()\*(UB(j)-LB(j)));**
3. end
4. **for iteration from 1 to** *I*
5. **for** each *i* ∈[N,1] **do,**
6. **for** each *j*∈[NV,1] **do,**
7. Velocity of swarm updated by
8. $V_{i,j}^{t+1} = wV_{i,j}^{t} + C_1 rand[0,1] \text{ X } (Pbest_{i,j}^{t} - X_{i,j}^{t}) + C_2 rand[0,1] \text{ X } (Gbest - X_{i,j}^{t})$
9. Update the position of swarms by:
10. $X_{i,j}^{t+1} = V_{i,j}^{t+1} + X_{i,j}^{t}$
11. Amend the $X_{i,j}^{t+1}$ between LB and UB
12. **end**
13. Evaluate the fitness of swarm
14. Update personal best by:
15. **if** $fitness(X_{i,j}^{t+1}) < fitness(Pbest_{i,j}^{t})$ **then,**
16. $Pbest_{i,j}^{t} = X_{i,j}^{t+1}$
17. **end**
18. Update global best by:
19. **if** $fitness(Pbest_{i,j}^{t}) < fitness(Gbest)$ **then**
20. $Gbest = Pbest_{i,j}^{t}$
21. **end**
22. **end**
23. Stopping criteria check. If iteration<*I* go to step 5, otherwise stop.
24. **end**

**Fig 9. The pseudocode of the proposed PSO algorithm.**

Fig 10 shows a comparison of the DC link voltages for PI controllers with and without optimisation technique. As can be seen, the rise time of the PI controller without optimisation is 0.1322 s, whilst that with PSO technique is 0.112 s. The reason is that the DC link voltage with optimisation technique responds faster than the conventional PI controller in stabilising the DC input voltage for the inverter system. At 0.4 s, the load step on the grid system suddenly changes. Moreover, the PI controller with the PSO technique has come down to a steady-state condition that is faster than the conventional PI controller. Therefore, the PSO technique can compensate the transient effect on the system to provide the stable DC input to the inverter system. The DC link input voltage is stabilised to ensure that a low ripple factor percentage is generated on the inverter AC output waveform.

The three-phase voltage and current outputs of the system with the controller are depicted in Figs 11 and 12, respectively. Fig 11 shows a comparison of the output voltages of the grid system obtained from simulations using PI controllers with and without the PSO technique.

Table 4. Comparison of the final parameters values with different technique.

| Item | PI Controller | | |
|---|---|---|---|
| | Trial and Error | Ziegler-Nichols | PSO Technique |
| Proportional gain of voltage regulator, $K_p^v$ | 0.532 | 2.05 | 1.4575 |
| Integral gain of voltage regulator, $K_i^v$ | 10 | 156 | 3.1055 |
| Proportional gain of active current controller, $K_p^d$ | 20 | 0.32 | 14.575 |
| Integral gain of active current controller, $K_i^d$ | 250 | 20 | 13.105 |
| Proportional gain reactive current controller, $K_p^q$ | 3.052 | 1.5 | 0.9226 |
| Integral gain of active current controller, $K_i^q$ | 150 | 20 | 16.349 |
| DC link oscillation peak | 620.47 | 599.8 | 532.12 |
| Frequency, Hz | 50 | 50 | 50 |

At t = 0.4–6 s, the load step on the grid system suddenly increases, and a slight noise is observed in the voltage amplitude from the conventional PI controller technique. Fig 11 shows that the output voltage from the conventional PI controller has harmonic waveform at 0.065–

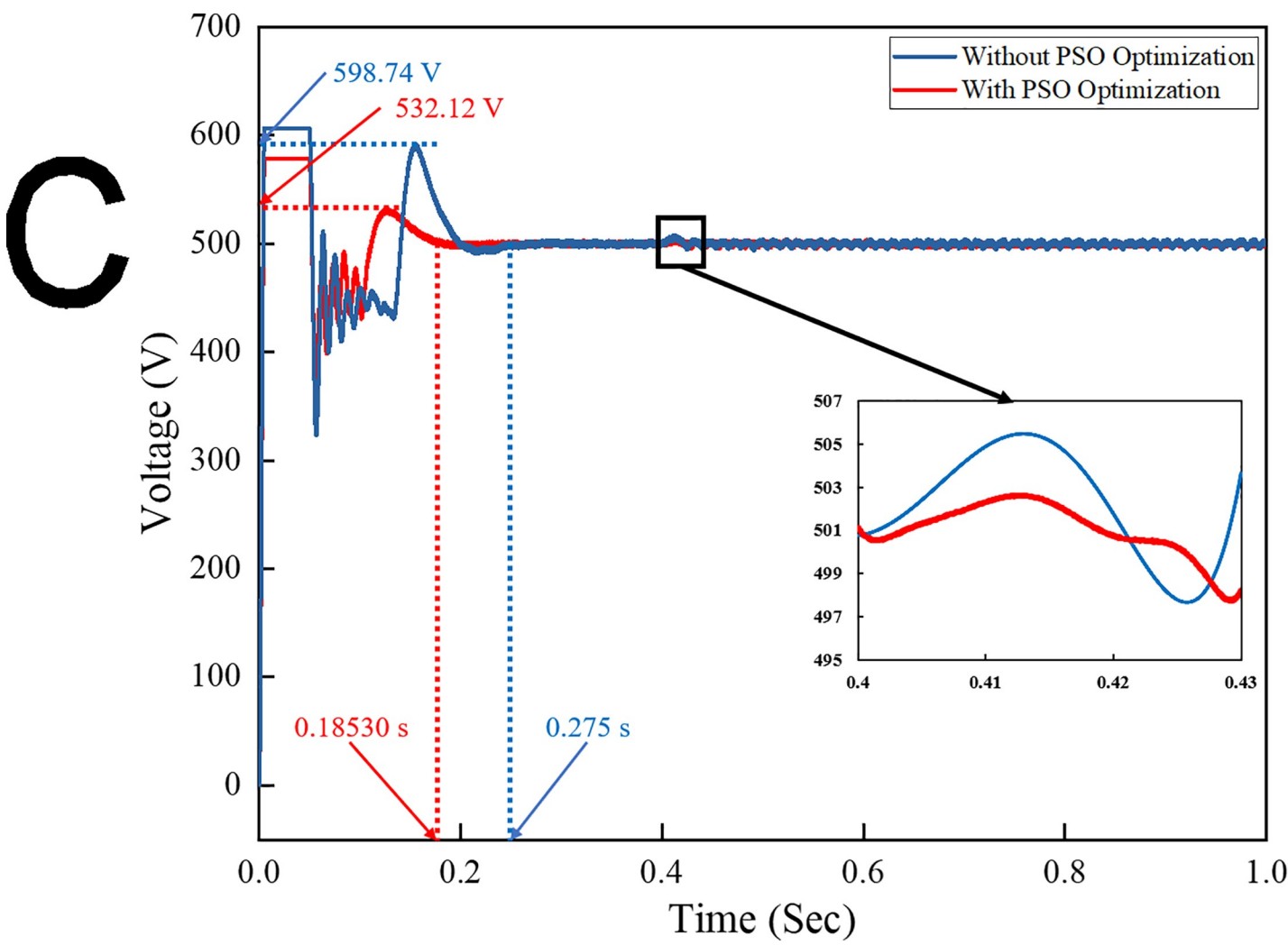

Fig 10. DC link voltage for both the conventional and optimization methods.

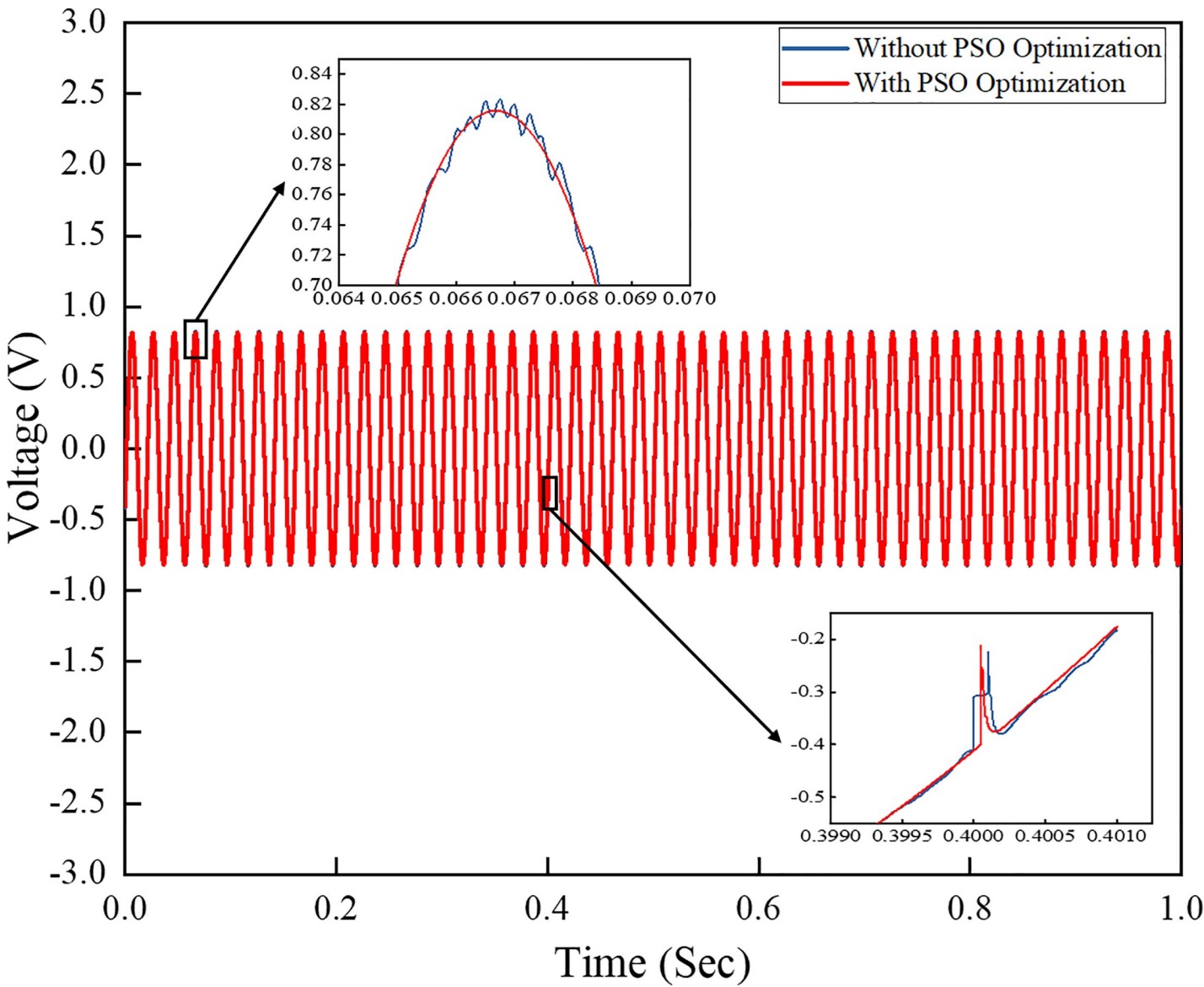

**Fig 11. Three-phase output voltage comparison of the output grid system.**

0.07s compared to the output voltage of the PI controller with PSO technique, which has less harmonic output waveform. This result is due to the proper control algorithm implemented in the inverter control scheme, which minimises the harmonic level in the output signal and provides a stable output voltage on the system. The amplitude is stable and fixed during the sudden load change.

Fig 12 shows a comparison of the output currents of the systems implementing a conventional PI controller and another PI controller with the PSO technique. The results show that the outputs vary depending on the sudden load step change occurring from 0.4–0.6 s. The currents vary with the load change increase on the system as the load draws more current from the grid system. The harmonic level on the output waveform of the PI controller without PSO is higher than the output waveform of the one with PSO technique. This finding proves that

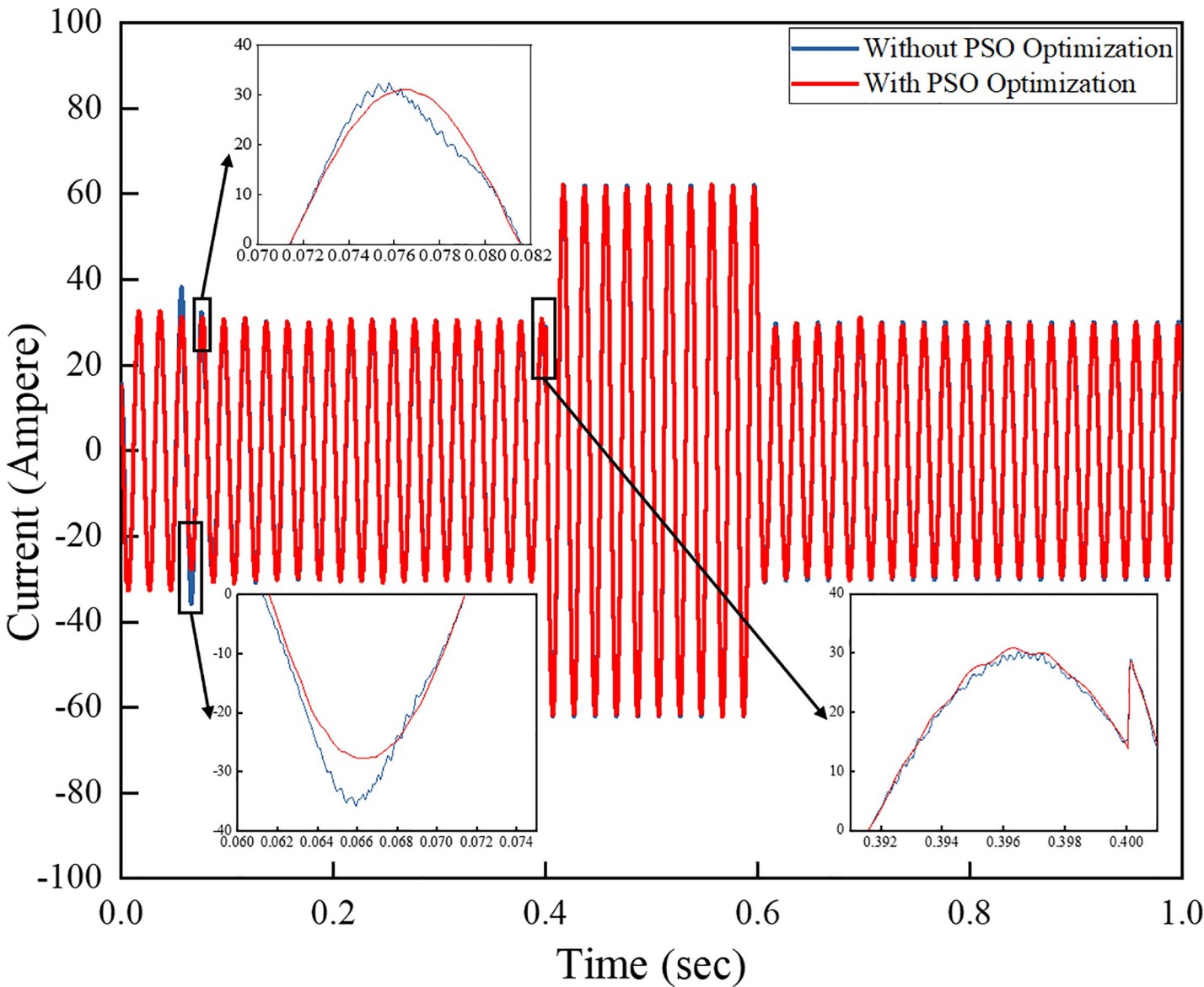

**Fig 12. Three-phase output current for the grid system.**

the controller effectively minimises the harmonic level on the output waveform whilst stabilising the amplitudes of voltage and current during the transient condition.

The power flow of the PV generation with a grid-connected inverter system along with a controller algorithm, as shown on Fig 13, should be analysed to support the load demand. From 0–0.4 s, the load of 100 kW is injected, and no power flow is applied to the grid as PV generation and load are balanced. From 0.4–0.6 s, the load power is increased to 30 kW. At this state, the load demand is higher than PV generation. Given that PV power generates a power of 100 kW, the additional power is drawn from the grid, which is indicated by –30 kW on the grid contribution profile line. The grid side continues to donate power until 0.6 s when the load is reduced to 40 kW. The low demand occurs, and the load then uses less power than it generates. Thus, the excess power of approximately 10 kW from the PV generation is dispatched to the grid, which is shown as 10 kW on the grid power profile line. The ability of the

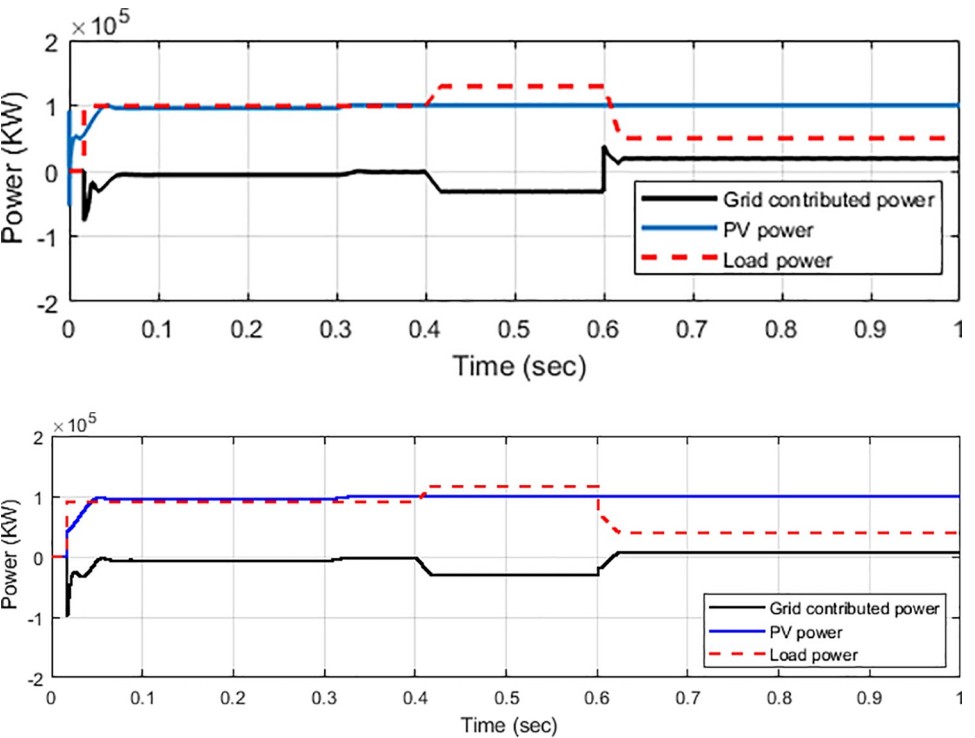

**Fig 13. Power flow analysis.** (a) PI controller and (b) PI with PSO technique.

inverter to extract power from PV generation and feed such power to the load demand and grid demonstrates the inverter's capacity for power flow analysis. As observed from the figure, the step transient condition occurs when load demand changes suddenly. The overshoot can be analysed as transient occurs. In general, a grid-connected system is in proportional transient condition to the flexibility power of solar radiations, frequency, and load variation. Fig 13 (A) shows that, compared with the PI controller with PSO technique, after using a conventional PI controller, the inverter system reveals much transient as well as overshoots in the PV and grid-contributed power as step transient condition occurs. Meanwhile, the PI controller with PSO technique mitigates the overall transient effects and minimises the overshoot, thus returning back to steady state condition, as shown in Fig 13(B). The PI controller with the PSO technique can compensate the transient effect and system overshoot when the load demand is increasing and decreasing, respectively.

The high-quality power from the system with low harmonic contents in terms of voltage and current is a crucial element in power system application, especially when connecting to the grid system. The reason is that high-quality output power with low THD factor on voltage and current signal provides better and linear power to the consumer, thus leading to energy savings. The THD factor is inversely proportional to the power factor of the loads. The load consuming less power factor yields a high THD factor to the grid system. By utilising an improved controller and fast Fourier transform analysis, the THD factor of the output phase voltage and current can be reduced to an acceptable level of less than 5%, which satisfies the IEEE Std. 929–2000 [39, 46].

Figs 14 and 15 present the harmonic spectrum of voltage and current output waveforms with conventional PI controller and PI controller with PSO technique, respectively. The results of the harmonic spectrum of voltage show 0.29% on the inverter output voltage with the

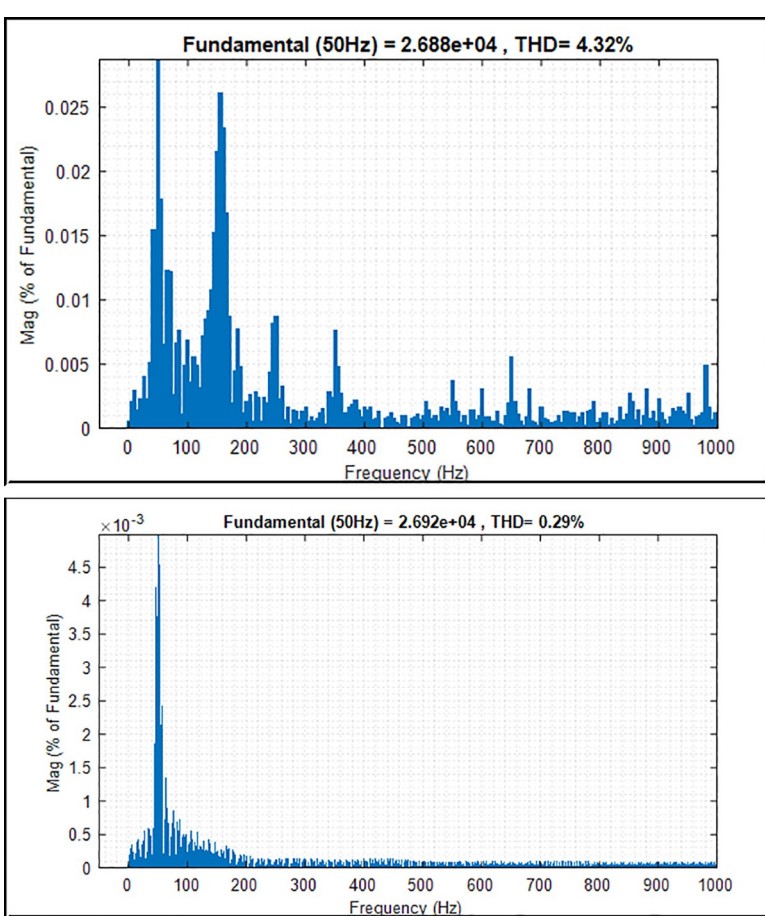

**Fig 14. THD and harmonic spectrum of the inverter output voltage.** (a) Conventional PI controller; (b) PI controller with PSO.

implemented PSO technique. This indicates a reduction of 4.32% compared with that of the conventional PI controller, as shown in Fig 14(A) and 14(B). Fig 15 shows the harmonic spectrum of the current on the output inverter system; as can be seen, the values are 4.49% and 2.72% with the conventional PI controller and PI controller with PSO technique, respectively. Thus, improving the controller algorithm of the inverter system leads to an efficient performance of the system. The harmonic level is attributed to the effectiveness of the proposed controller, which implements the PSO technique to the inverter control system with voltage and current controllers, filter, and sinusoidal PWM technique.

Fig 16 shows that the PSO algorithm reduces the fitness function and converges the value at 48.6 iterations to reach the best result as compared to the Binary Coded Extremal Optimization (BCEO) algorithm, which converges at 68.9 iterations. The optimization has been run for 100 iterations to determine the best values of the PI controller parameters $K_p$ and $K_i$ obtained by using the PSO algorithm. This PSO is implemented to find the optimum values for the PI controller parameters for the voltage regulator and current controllers in the three-phase inverter system. As indicated by the results, compared with trial-and-error technique, the PSO technique has good performance in the controller compared with that using BCEO algorithm in terms of finding the best values for the PI controller.

Fig 17 shows the frequency response of the grid-connected PV system with inverter control algorithm. The frequency shows that the load demand at 0.4 s is increased and has drawn

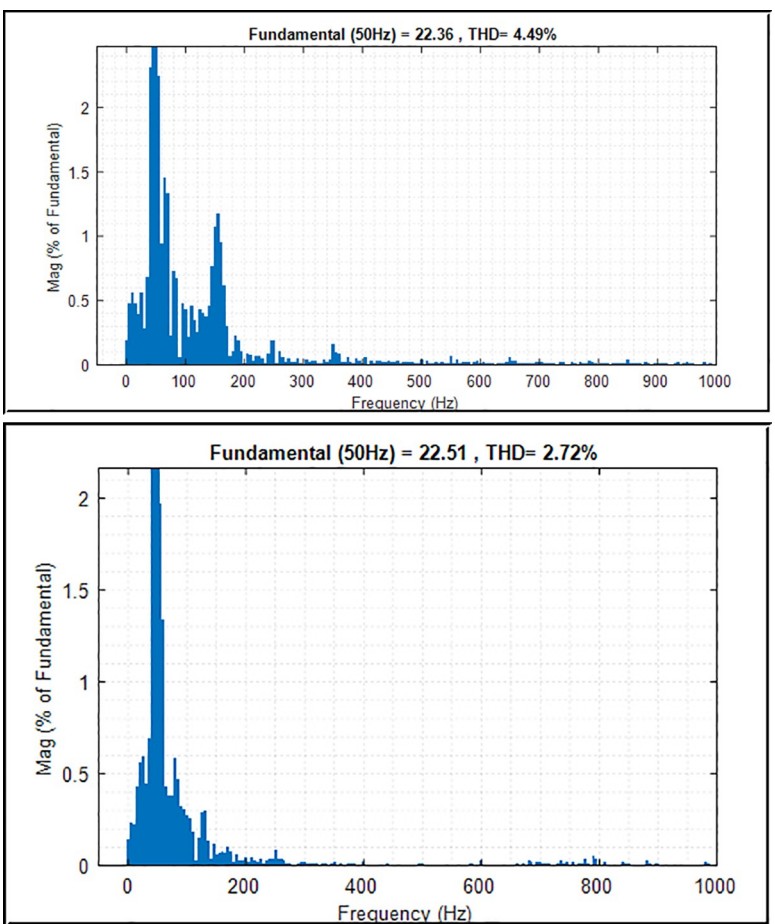

**Fig 15. THD and harmonic spectrum of the inverter output current.** (a) Conventional PI controller; (b) PI controller with PSO.

more current from the grid to support the load demand. At 0.6 s, the load demand is suddenly reduced, and as a result, the current is injected back to the grid to provide a stable power supply to the load as the load demand decreases. The frequency responses based on the PI controller with PSO and PI controller without optimisation technique are compared. The frequency responses of the PI controller with PSO technique are much stable and constant during load demand changes, because the PSO algorithm in the controller compensates for the overshoot during any changes.

As previously mentioned, the active current references $i_d^*$ in the $d-q$ axis reference frame is determined by the voltage regulator controller block and is obtained from the output of the PI controller with PSO technique. Fig 18 shows a comparison of the $i_d^*$ output curves from the conventional PI controller and PI controller with PSO technique. The active and reactive powers injected on the grid can be controlled by controlling $i_d^*$ and the reactive current reference $i_q^*$, respectively. The $i_d^*$ reference outputs from the PI controller with PSO are more stable and constant than those obtained from the conventional PI controller.

Moreover, the performance of $i_d^*$ can also be evaluated based on the disturbance lines occurring in the utility grid system. Fig 19 shows the performance responses of $i_d^*$ in which grid disturbance occurred at 0.7 s. The grid side voltage and power are not available at this time due to the disturbance that occurred at the transmission line system. It can be seen that the $i_d^*$

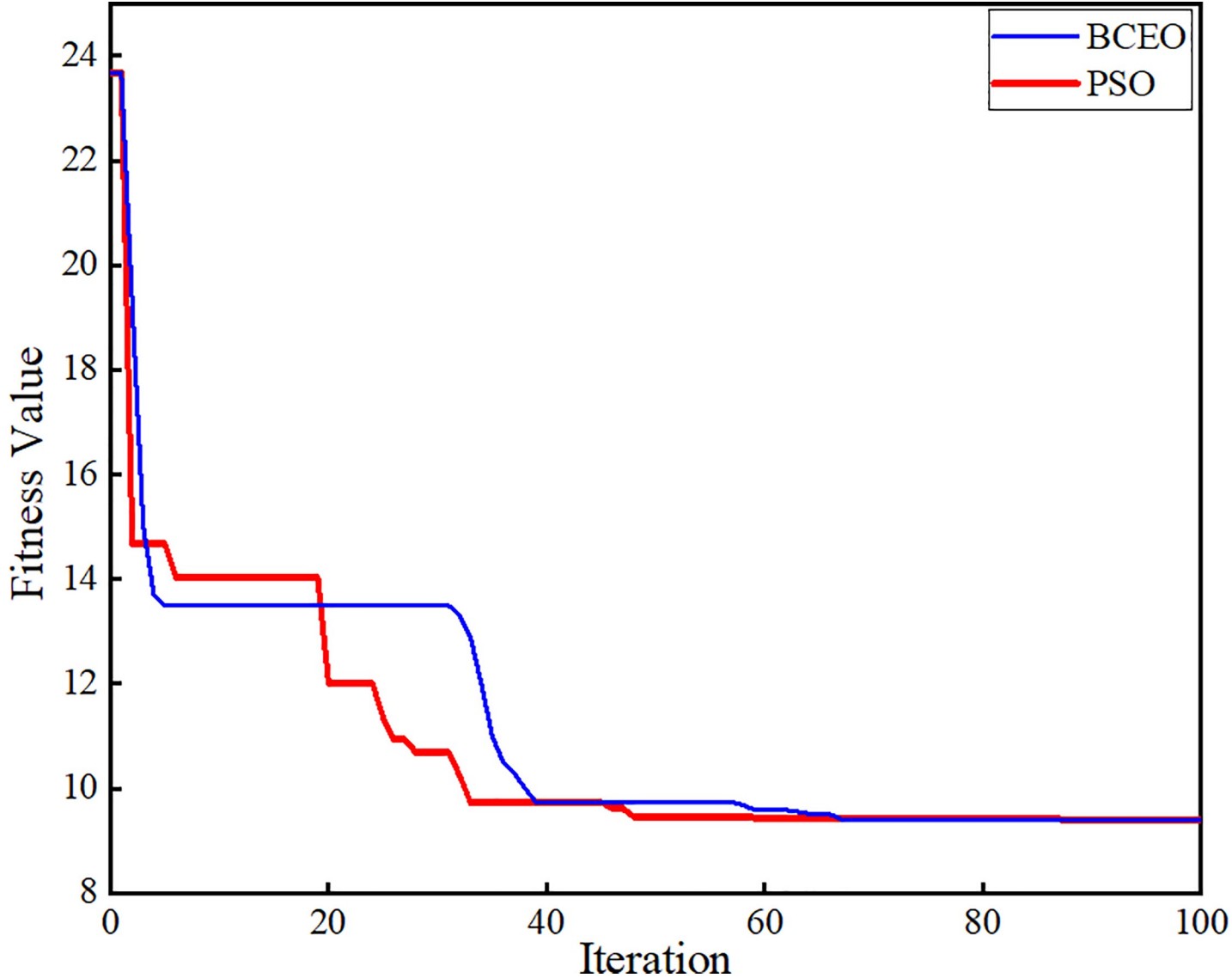

**Fig 16. Relationship curve between fitness function and iteration.**

reference responses from the PI controller with PSO optimization technique are highly controllable and stable during a disturbance on the utility grid side. This is because the PSO optimization technique is able to cater and synchronize the sudden disturbance occurred by properly tuning and adjusting the PI controller parameter to obtain an optimum value for the system. Hence, the PSO optimization technique is robust and can effectively control the PI controller in the grid-connected three phase PV inverter system, thus providing a stable inverter system output.

The performance comparison of the conventional PI controller and the one using the PSO technique is presented in Table 5. In this table, the comparison with other reported works is performed based on the Ziegler–Nichols method, the PI-PSO method, the binary coded extremal optimization (BCEO), and the PSO algorithm optimization. The Ziegler–Nichols method is the most well-known technique in the control system. In addition, a previous study [40] demonstrated that the conventional Ziegler–Nichols technique yields better performance than

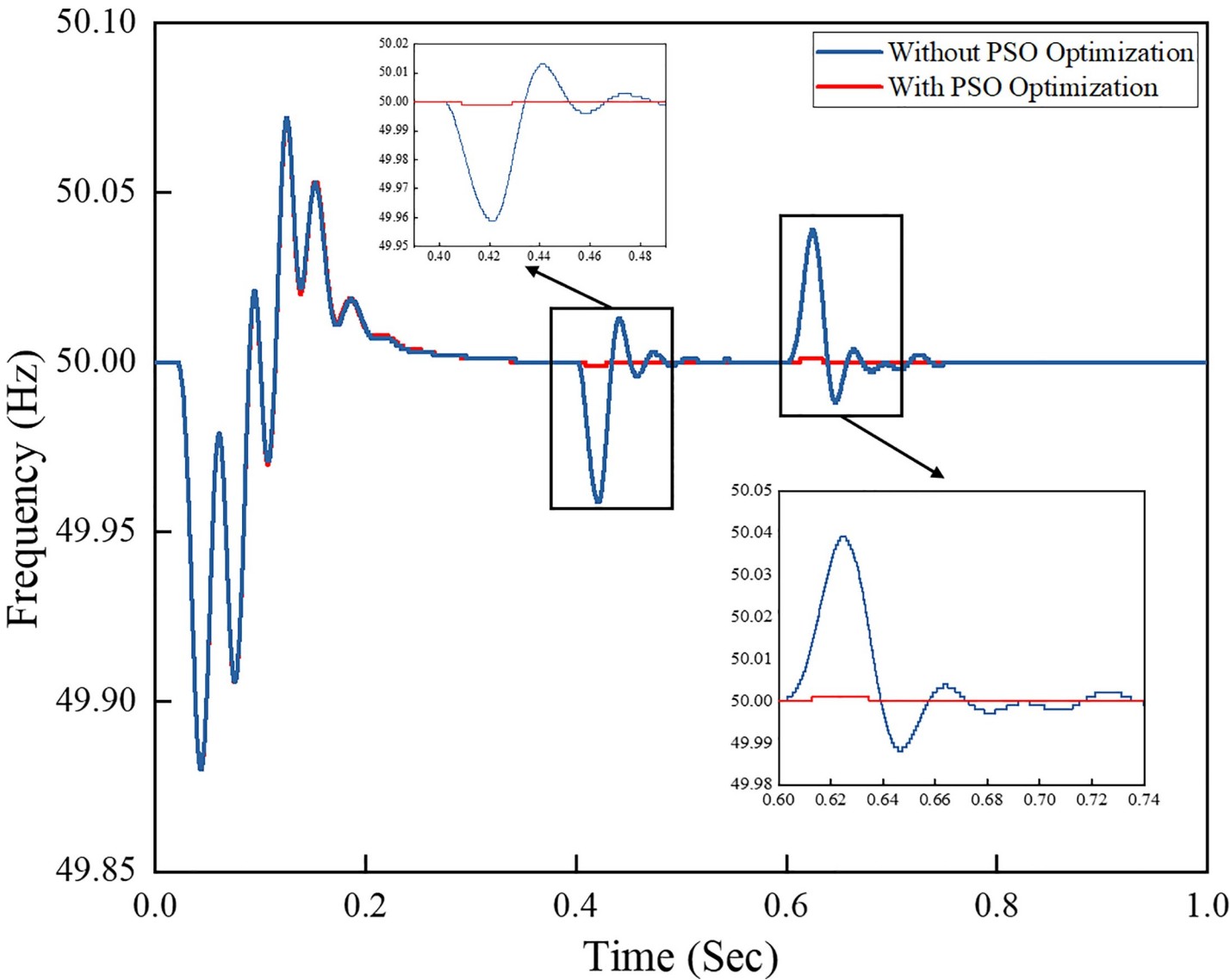

**Fig 17. Frequency response of the grid-connected PV system.**

the fuzzy logic controller. The conventional PI controller technique is thus chosen based on the Ziegler–Nichols method. The optimum value of the PI controller is determined based on the ultimate gain $K_u$ and ultimate period $T_u$ of the feedback control system in the voltage control loop and the inner control loop. The ultimate gain and ultimate period of oscillation are used to represent the dynamic characteristic of the process. By assuming $K_i$ as zero, the value of $K_p$ and the ultimate period $T_u$ can be determined.

By implementing a Ziegler–Nichols table for the PI controller, the value of such a controller can be determined. Table 5 summarizes the percentage of improvement of implementing the PSO algorithm in the system. As shown in the table, the proposed strategy outperformed other reported techniques. Using the PSO algorithm in this work, the results indicated that the THD for grid voltage and current are 0.29% and 2.72%, respectively, which are lower than those reported in other works. The improvement of time to reach steady state condition and rise time are much faster than others, which are 0.1853 and 0.112 s respectively. The two main

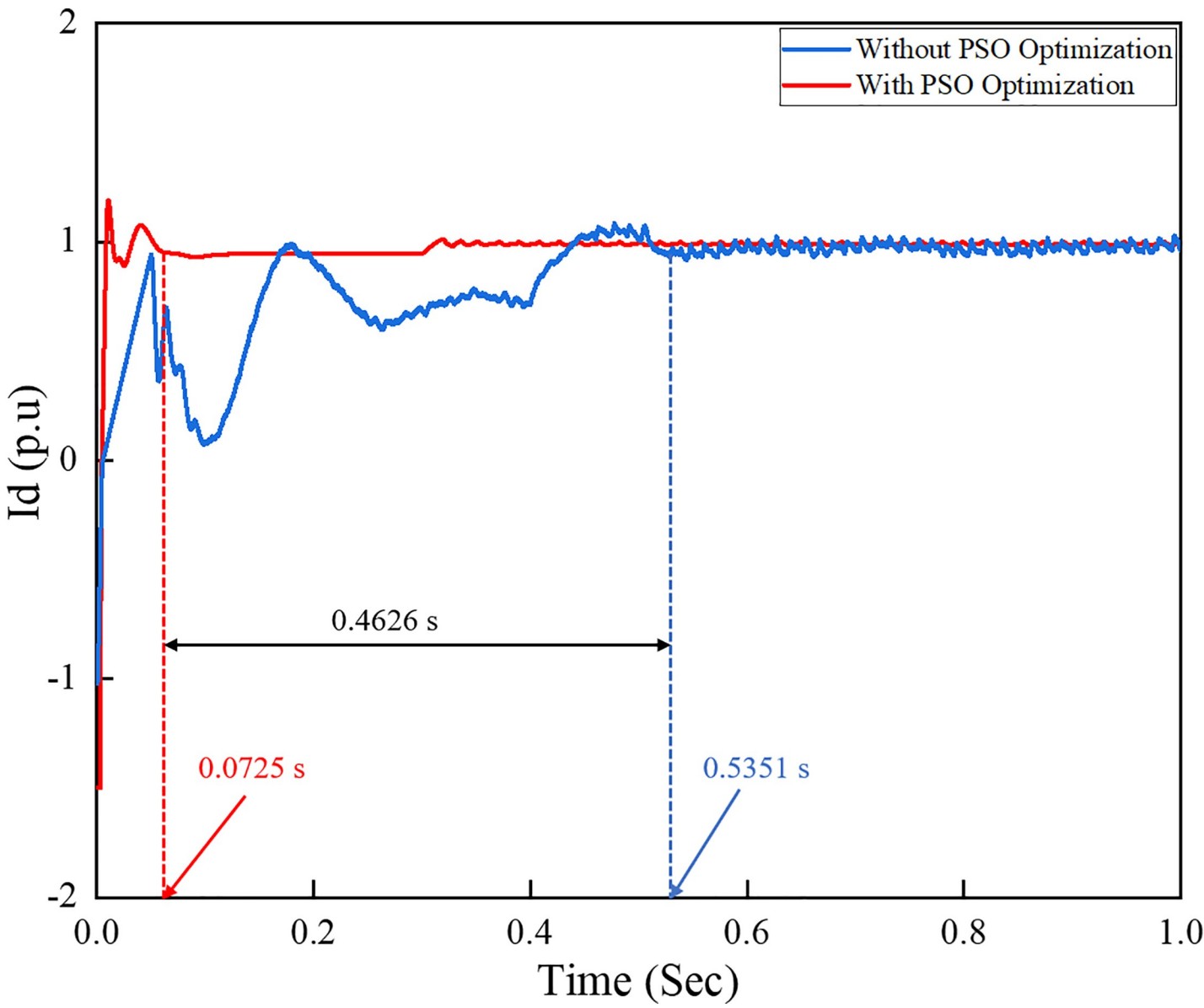

**Fig 18.** Active current references $i_d^*$ of the inverter control system under load variation.

factors that contribute to the significant improvement in grid performance are the additional input parameters and the optimized input parameters of the control system. Compared to other works that only utilized either the voltage control loop or the current control loop, the PSO algorithm in the current work considers the voltage control loop, the reactive power control loop, and the current control loop with feed-forward decoupling method. All the PI controllers involved in the input parameters were optimized by the PSO algorithm. The optimized input parameters of the controller were taken as error and change of error. The PSO algorithm instantaneously tracks the error and change of error based on load variation and grid disturbance and provides the optimum values to the controller. Due to the additional input parameters and constraints considered in the optimization, the computational times taken for PSO algorithm to simulate the model and track the error and change of error are longer. However,

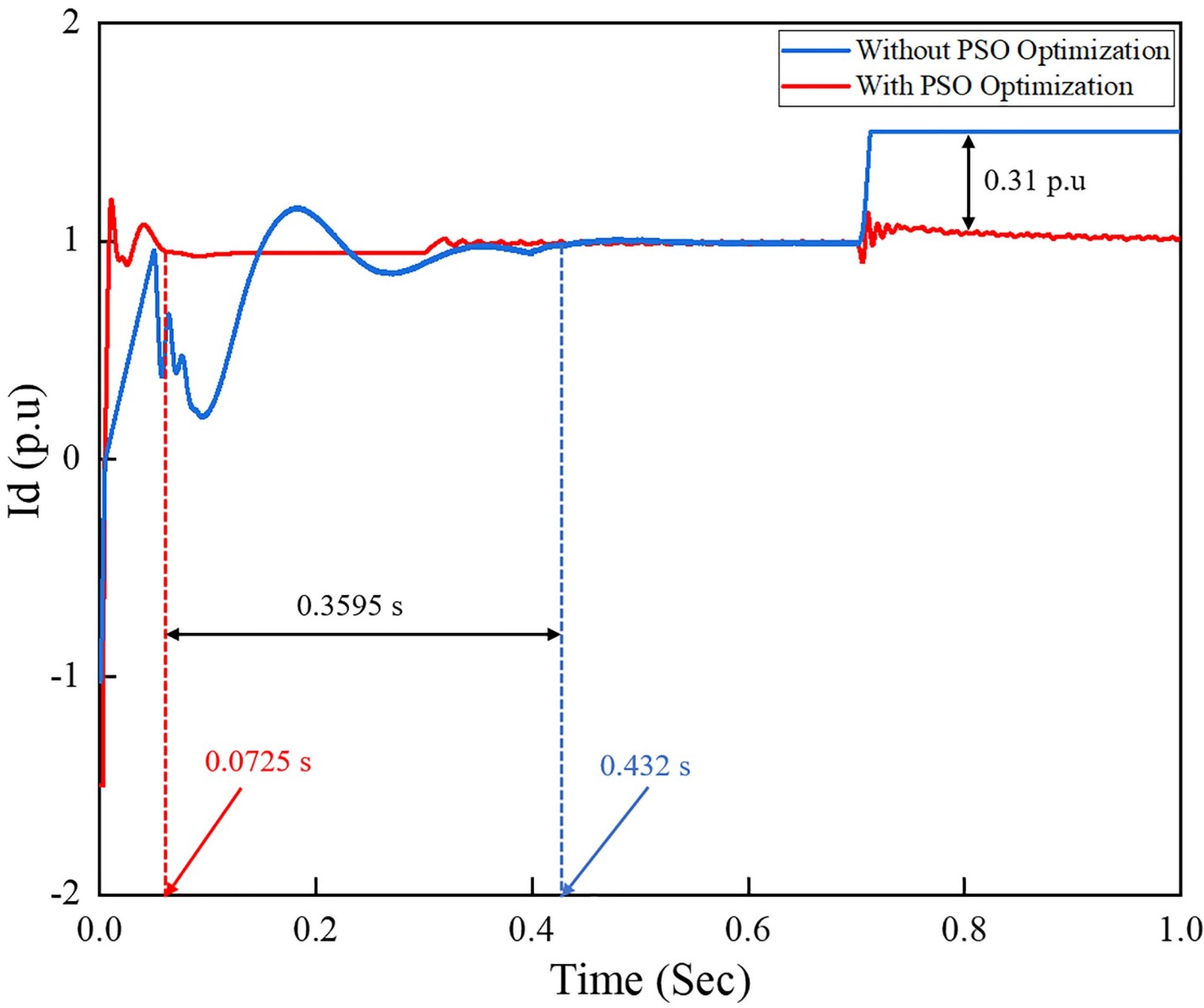

**Fig 19.** Active current references $i_d^*$ of the inverter control system under grid disturbance.

the rise time taken for the controller to initiate the error in the system is much faster compared to other reported works and provides the optimum result and better performance to rapidly track the reference active and reactive current of the inverter system.

## 7. Conclusion

The modelling and testing of a proposed controller for the high-performance control of a three-phase grid-connected PV system is presented in this paper. The system is simulated and tested in the MATLAB/Simulink environment. The PSO technique is implemented to obtain the optimal values of the PI controller parameters by minimising the error of voltage regulator and current controllers. The technique is also used to minimize the error as much as possible

**Table 5. Comparison of reported works.**

| Item | PI Controller | | | | |
|---|---|---|---|---|---|
| | Ziegler–Nichols [5] | PI-PSO Technique [47] | BCEO-PI controller [48] | PSO Algorithm | Percentage of Improvement from [5, 47, 48] (%) |
| Total harmonic distortion (%), THD of voltage | 4.32 | 4.20 | 2.41 | **0.29** | **93.2; 93; 87.9** |
| Total harmonic distortion (%), THD of current | 4.49 | 2.9261 | N/A | **2.72** | **39.4; 7.04; N/A** |
| Time to reach steady state condition (sec) | 0.275 | 0.38 | 0.2356 | **0.1853** | **32.6; 51.2; 21.3** |
| Frequency, (Hz) | 50 | 50 | 50 | **50** | **N/A** |
| Rise time (sec) | 0.223 | 0.3441 | 0.1652 | **0.112** | **49.7; 67.4; 32.2** |

and find the best parameters of the PI controller. The effects of load transient to the grid system as well as grid disturbance are also discussed.

The developed PSO technique shows the best responses to the inverter controller in determining the optimum values for the PI controller and minimizing the error to reduce overshoot, transient response, and steady-state error. The results also shows that, under load variation, the THD percentages for both the voltage and current of the inverter output signal are much lower compared at 0.29% and 2.72%, respectively, compared to other reported works. Furthermore, the time to reach steady state condition for input voltage is much lower (0.1853 s) than others, indicating an over 20% faster response under a normal frequency of 50 Hz.

Finally, the rise time taken for DC link to respond to the fluctuation of the system is 0.112 s, which is over 30% faster compared to other reported results. These findings reveal that the proposed strategy outperforms other research works by introducing additional relevant input parameters and optimizing such input parameters. In utilizing this technique, the proposed strategy shows a better performance with regard to the DC link fluctuation, voltage and current stabilisation, harmonics reduction, and frequency stability. The developed robust controller can be used for the inverter controller system in power system applications, thus ensuring good performance enhancement and power quality improvement.

The intermittency of output generation by the PV system based on sun irradiation leads to unstable power supply to the loads especially where utility is unavailable. Thus, improper control and optimal design of controller leads to pure of power quality and stability performance of the inverter system. Therefore, in this study the PSO algorithm is implemented in the PI controller to obtain an optimum values of PI controller parameters to improve system performance in real time simulation system. In addition, a real time experimental setup of the system has been considered as future work of this system.

## Author Contributions

**Formal analysis:** M. F. Roslan, A. W. M. Zuhdi.

**Methodology:** Ali Q. Al-Shetwi.

**Software:** M. F. Roslan.

**Visualization:** A. W. M. Zuhdi.

**Writing – original draft:** M. F. Roslan.

**Writing – review & editing:** M. F. Roslan, M. A. Hannan, P. J. Ker.

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
