## [Decision Letter · Decision Letter 0]

21 Sep 2020

PONE-D-20-26206

Particle Swarm Optimization Algorithm-Based PI Inverter Controller for Grid-Connected PV System

PLOS ONE

Dear Dr. Roslan,

Thank you for submitting your manuscript to PLOS ONE. After careful consideration, we feel that it has merit but does not fully meet PLOS ONE’s publication criteria as it currently stands. Therefore, we invite you to submit a revised version of the manuscript that addresses the points raised during the review process.

The authors should revise their paper carefully according to the reviewers' comments.

We look forward to receiving your revised manuscript.

Kind regards,

Long Wang, Ph.D.

Academic Editor

PLOS ONE

Journal Requirements:

"This work was funded by the Ministry of Higher Education, Malaysia under Universiti Tenaga Nasional using grant no. 20190101LRGS."

5. Please upload a new copies of Figures 5 and 9 as the detail are not clear. Please follow the link for more information: https://blogs.plos.org/plos/2019/06/looking-good-tips-for-creating-your-plos-figures-graphics/

Reviewers' comments:

Reviewer's Responses to Questions

**Comments to the Author**

1. Is the manuscript technically sound, and do the data support the conclusions?

Reviewer #1: Yes

Reviewer #2: Yes

2. Has the statistical analysis been performed appropriately and rigorously? 

Reviewer #1: Yes

Reviewer #2: Yes

3. Have the authors made all data underlying the findings in their manuscript fully available?

Reviewer #1: Yes

Reviewer #2: Yes

4. Is the manuscript presented in an intelligible fashion and written in standard English?

Reviewer #1: Yes

Reviewer #2: Yes

5. Review Comments to the Author

Reviewer #1: Authors proposed the performance of a control strategy for an inverter in a three-phase grid-connected PV system. There are some concerns, as listed below.

1) The authors are recommended to highlight the novelty and contribution of the work.

2) The authors are recommended to highlight the advantages of their proposed method in comparison with conventional methods.

3) How the proposed method has improved the performance evaluation compared with JAYA algorithm [1]?

[1] Wang L, Huang C, Huang L. Parameter estimation of the soil water retention curve model with Jaya algorithm[J]. Computers and Electronics in Agriculture, 2018, 151: 349-353.

4) Authors should include references from 2019 to 2020.

Reviewer #2: Q.1 How to design an effective and efficient PI controller for a three-phase grid-connected inverter to obtain satisfied power quality performance is widely known research topics. Only the PI controller design for simulation model is not enough. Authors should be added new section that is experimental section if funds allow.

Q.2 In a grid-connected photovoltaic system, the frequency and the harmonics of voltage is determined by the larger grid, but power quality, stability and power mismatch can be improved by the three-phase grid-connected inverter, so the author needs to do some more jobs about power quality.

Q.3 This manuscript does not bring any new knowledge on evolutionary algorithm, so the innovations focus on novel applications. The evolutionary algorithm, such as BCEO ,PSO, should be implemented in the novel model in this manuscript, and the convergence process of PSO and BCEO should be shown in Fig.16.

6. PLOS authors have the option to publish the peer review history of their article (what does this mean?). If published, this will include your full peer review and any attached files.

Reviewer #1: No

Reviewer #2: No

---

## [Author Response · Author response to Decision Letter 0]

25 Oct 2020

Thank you for your time in reviewing our paper and look forward to meeting your expectations. Since your inputs have been precious, in the eventuality of a publication, we would like to acknowledge your contribution explicitly. The authors

---

## [Decision Letter · Decision Letter 1]

24 Nov 2020

Particle Swarm Optimization Algorithm-Based PI Inverter Controller for a Grid-Connected PV System

PONE-D-20-26206R1

Dear Dr. Roslan,

We’re pleased to inform you that your manuscript has been judged scientifically suitable for publication and will be formally accepted for publication once it meets all outstanding technical requirements.

Kind regards,

Long Wang, Ph.D.

Academic Editor

PLOS ONE

Additional Editor Comments (optional):

Reviewers' comments:

Reviewer's Responses to Questions

**Comments to the Author**

1. If the authors have adequately addressed your comments raised in a previous round of review and you feel that this manuscript is now acceptable for publication, you may indicate that here to bypass the “Comments to the Author” section, enter your conflict of interest statement in the “Confidential to Editor” section, and submit your "Accept" recommendation.

Reviewer #1: All comments have been addressed

Reviewer #2: All comments have been addressed

2. Is the manuscript technically sound, and do the data support the conclusions?

Reviewer #1: Yes

Reviewer #2: Yes

3. Has the statistical analysis been performed appropriately and rigorously? 

Reviewer #1: Yes

Reviewer #2: Yes

4. Have the authors made all data underlying the findings in their manuscript fully available?

Reviewer #1: Yes

Reviewer #2: Yes

5. Is the manuscript presented in an intelligible fashion and written in standard English?

Reviewer #1: Yes

Reviewer #2: Yes

6. Review Comments to the Author

Reviewer #1: Good work! I'm very happy that all my concerns have been addressed.

Reviewer #2: The authors have made sufficient modifications according to the modification comments, and I suggest that this paper be accepted only if the following issues are improved.

Q.1 There are some errors in the manuscript, such as, in page 63, Table 2, Line Impedance should be Line resistance and XLg should be Xg.

Q.2 The convergence process of PSO and BCEO are compared in Fig.16, but the number of iterations should be the same in Fig.16.

7. PLOS authors have the option to publish the peer review history of their article (what does this mean?). If published, this will include your full peer review and any attached files.

Reviewer #1: No

Reviewer #2: No

---

## [Editor Report · Acceptance letter]

1 Dec 2020

PONE-D-20-26206R1 

Particle Swarm Optimization Algorithm-Based PI Inverter Controller for a Grid-Connected PV System 

Dear Dr. Roslan:

I'm pleased to inform you that your manuscript has been deemed suitable for publication in PLOS ONE. Congratulations! Your manuscript is now with our production department. 

Kind regards, 

on behalf of

Dr. Long Wang 

Academic Editor

PLOS ONE